# TensorVAE: a simple and efficient generative model for conditional molecular conformation generation

**Hongyang Yu** *hyyu@anticancerbio.com*
*Anticancer Bioscience (ACB), Ltd., Sydney, Australia*

**Hongjiang Yu** *hjyu@anticancerbio.com*
*Anticancer Bioscience (ACB), Ltd., Sydney Australia*

**Reviewed on OpenReview:** *https://openreview.net/forum?id=rQqzt4gYcc*

## Abstract

Efficient generation of 3D conformations of a molecule from its 2D graph is a key challenge in in-silico drug discovery. Deep learning (DL) based generative modelling has recently become a potent tool to tackling this challenge. However, many existing DL-based methods are either indirect–leveraging inter-atomic distances or direct–but requiring numerous sampling steps to generate conformations. In this work, we propose a simple model abbreviated TensorVAE capable of generating conformations directly from a 2D molecular graph in a single step. The main novelty of the proposed method is *focused on feature engineering*. We develop a novel encoding and feature extraction mechanism relying solely on *standard convolution operation* to generate token-like feature vector for each atom. These feature vectors are then transformed through *standard transformer encoders* under a conditional Variational Autoencoder framework for generating conformations directly. We show through experiments on two benchmark datasets that with intuitive feature engineering, a relatively simple and standard model can provide promising generative capability outperforming more than a dozen state-of-the-art models employing more sophisticated and specialized generative architecture. Code is available at `https://github.com/yuh8/TensorVAE`.

## 1 Introduction

Recent advance in deep learning has enabled significant progress in computational drug design (Chen et al., 2018). Particularly, capable graph-based generative models have been proposed to generate valid 2D graph representation of novel drug-like molecules (Honda et al., 2019; Mahmood et al., 2021; Yu & Yu, 2022), and there is an increasing interest on extending these methods to generating 3D molecular structures which are essential for structured-based drug discovery (Li et al., 2021; Simm et al., 2021; Gebauer et al., 2022). A stable 3D structure or conformation of a molecule is specified by the 3D Cartesian coordinates of all its atoms. Traditional molecular dynamics or statistical mechanic driven Monte Carlo methods are computationally expensive, making them unviable for generating 3d molecular structures at scale (Hawkins, 2017). In this regard, deep learning(DL)-based generative methods have become an attractive alternative.

DL-based generative methods may be broadly classified into 4 categories: distance-based, reconstruction-based, sequential and energy-based and direct methods. The main goal of distance-based methods is learning a probability distribution over the inter-atomic distances. During inference, distance matrices are sampled from the learned distribution and converted to valid 3D conformations through post-processing algorithms. Two representative methods of this category include GraphDG (Simm & Hernández-Lobato, 2020) and CGCF (Xu et al., 2021a). An advantage of modeling distance is its roto-translation invariance property–an important inductive bias for molecular geometry modeling (Köhler et al., 2020). Additional virtual edges and their distances between $2^{nd}$ and $3^{rd}$ neighbors are often introduced to constrain bond angles

and dihedral angles crucial to generating a valid conformation. However, Luo et al. (2021) have argued that these additional bonds are still inadequate to capture structural relationship between distant atoms. To alleviate this issue, DGSM (Luo et al., 2021) proposed to add higher-order virtual bonds between atoms in an expanded neighborhood region. Another weakness of the distance-based methods is the error accumulation problem; random noise in the predicted distance can be exacerbated by an Euclidean Distance Geometry algorithm, leading to generation of inaccurate conformations (Xu et al., 2022; 2021b).

To address the above weaknesses, reconstruction-based methods directly model a distribution over 3D coordinates. Their main idea is to reconstruct valid conformations from distorted coordinates. GeoDiff (Xu et al., 2022) and Uni-Mol (Zhou et al., 2023) are pioneering studies in this respect. Though sharing similar idea, they differ in the process of transforming corrupted coordinates to stable conformations. While GeoDiff adapts a reverse diffusion process (Sohl-Dickstein et al., 2015), Uni-Mol treats conformation reconstruction as an optimization problem. Despite their promising performance, both methods require designing of task-specific and complex coordinate transformation methods. This is to ensure the transformation is roto-translation or SE(3)-equivariant. To achieve this, GeoDiff proposed a specialized SE(3)-equivariant Markov transition kernel. On the other hand, Uni-Mol accomplished the same by combining a task-specific adaption of transformer (Vaswani et al., 2017) inspired by the AlphaFold's Evoformer (Jumper et al., 2021) with another specialized equivariant prediction head (Satorras et al., 2021). Furthermore, GeoDiff requires numerous diffusing steps to attain satisfactory generative performance which can be time consuming.

While promising generative performance has been achieved by directly learning a distribution over the 3D geometries (coordinates or pair-wise distances) of molecules, energy-based learning methods have also recently been shown to yield competitive performance in molecular conformation generation. A unique advantage of using energy minimization as the reward mechanism is that energy-based models can better explore the low-energy regions of the conformational space of a molecule, leading to generating conformations with both high quality and diversity. On the other hand, methods relying directly on minimizing distance metrics to ground-truth conformations may result in generating very similar conformations with high energy strain. Two recent methods adopting the energy minimization paradigm are TorsionNet (Gogineni et al., 2020) and GFlowNet (Volokhova et al., 2023) for conformation generation. Both of which are sequential and energy-based methods that sequentially move an original molecular conformation by modifying the torsion angle of all rotatable bonds towards a lower energy state. Despite their promising potential and advantages, sequential methods are relatively inefficient compared to the direct methods. For instance, GFlowNet requires iterating through 40,000 training steps per molecule to achieve satisfactory performance.

CVGAE (Mansimov et al., 2019) and DMCG (Zhu et al., 2022) have attempted to resolve the generative efficiency issue by developing models that can produce a valid conformation directly from a 2D molecular graph in a single sampling step. Regrettably, the performance of CVGAE is significantly worse than its distance-based counterparts mainly due to the use of inferior graph neural network for information aggregation (Zhu et al., 2022). DMCG aimed to improve the performance of its predecessor by using a more sophisticated graph neural network and a loss function invariant to symmetric permutation of molecular substructures. Although DMCG achieved superior performance, acquiring such loss function requires enumerating all permutations of a molecular graph, which can become computationally expensive for long-sequence molecules.

Regardless of their category, a common recipe of success for these models can be distilled to developing model architecture with ever increasing sophistication and complexity. There is little attention on input feature engineering. In this work, we forgo building specialized model architecture but instead focus on intuitive input feature engineering. We propose to encode a molecular graph using a fully-connected and symmetric tensor. For preliminary information aggregation, we run a rectangle kernel filter through the tensor in a 1D convolution manner. This operation has a profound implication; with a filter width of 3, the information from two immediate neighbors as well as all their connected atoms can be aggregated onto the focal atom in a single operation. It also generates token-like feature vector per atom which can be directly consumed by a standard transformer encoder for further information aggregation.

The generative framework follows the standard conditional variational autoencoder (CVAE) setup. We start with building two input tensors with one encoding only the 2D molecular graph and the other also encoding 3D coordinate and distance. Both tensors go through the same feature engineering step and the generated

feature vectors are fed through two separate transformer encoders. The output of these two encoders are then combined in an intuitive way to form the input for another transformer encoder for generating conformation directly. The complete generative model is abbreviated as TensorVAE.

In summary, the proposed method has 4 main advantages. (1) **Direct and Efficient**, generating conformation direclty from a 2D molecular graph in a single step. (2) **Simple**, not requiring task-sepecific design of neural network architecture, relying only on simple convolution and off-the-shelf transformer architecture; (3) **Easy to implement**, no custom module required as both `PyTorch` and `TensorFlow` offer ready-to-use convolution and transformer implementation. These advantages translate directly to excellent practicality of the TensorVAE method. (4) **Achieving competitive performance through simplicity**, we demonstrate through extensive experiments on two benchmark datasets that the proposed TensorVAE, despite its simplicity, can perform competitively against **22 recent state-of-the-art methods** for conformation generation and molecular property prediction.

## 2 Method

### 2.1 Preliminaries

**Problem Definition.** We formulate molecular conformation generation as a conditional generation task. Given a set of molecular graphs $G$ and their corresponding i.i.d conformations $R$, the goal is to train a generative model that approximates the Boltzman distribution, and from which a valid conformation conditioned on a molecular graph can be easily sampled in a single step.

**Story Line.** In the ensuing sections, we breakdown the formulation of the proposed method in three novel ideas. We first introduce how a molecular graph can be encoded using a 3D tensor. Then, we demonstrate how token-like feature vector can be generated from the input tensor by using a 1D convolution operation. The generated feature tokens resemble those used in the language modelling, thereby allowing the use of standard transformer encoders for effective information aggregation. Finally, we propose a novel mechanism to combine the outputs of the transformer encoders under a conditional-VAE framework to arrive at the final generative model.

### 2.2 Input tensor graph

Message passing graph neural network (GNN) is a popular feature extraction backbone for DL-based molecular conformation generation. The input for this backbone is often composed of three components, including atom features, edge features and an adjacency matrix. Atom and edge features normally pass through separated embedding steps before being fed to the GNN. Adjacency matrix is then used to determine neighboring atoms for layer-wise information aggregation. Although bond features are aggregated onto atom features and vice versa, these two features are maintained separately throughout the message passing layers (Gilmer et al., 2017; Satorras et al., 2021). Instead of having separated inputs, **our first simple idea** is to combine them into a single input. Specifically, we add an additional dimension to the adjacency matrix, making it a 3D tensor. Each cell on-diagonal of the tensor holds the *focal* atom feature vector to which information from nearby connected atoms are aggregated.

We consider three types of atom features comprising atom type, charge and chirality. Each feature is one-hot encoded and they are stacked together to form a single atom feature vector. There are two variants of the atom feature vector corresponding to two input tensors for the two encoders of the CVAE framework: an encoder conditioned only on graph (referred to as the $G$ tensor) and the other conditioned on both graph and coordinates (referred to as the $GDR$ tensor). For the GDR tenosr, every focal atom feature vector has three additional channels incorporating the 3D coordinate of the respective atom, and a distance channel filled with zeros.

Each off-diagonal cell holds the **stacked** *neighbour* atom and bond features. The considered bond features are bond type, bond stereo-chemistry type, ring size and normalized bond length. A virtual bond is also included in the bond type. It is worth noting that all virtual bonds share the same virtual bond type; they only differ in their normalized bond length. The normalized bond length is calculated as edge length (1 for

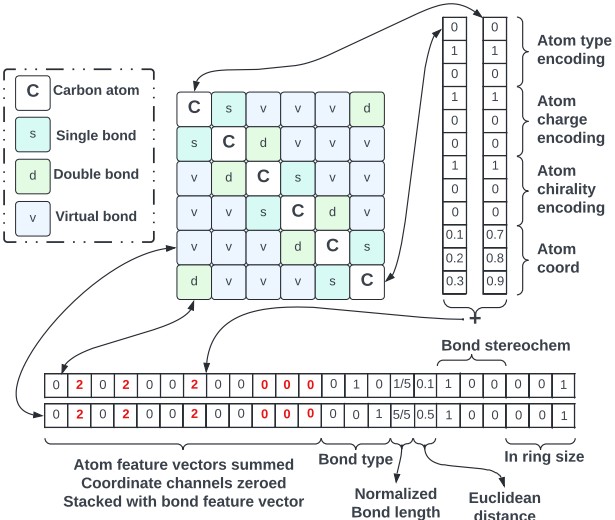

Figure 1: Benzene ring tensor graph example. Note that the values in the feature vector and its dimension are for demonstration purpose only. We explain how they are determined in Sec.3.

direct neighbor, 2 for $2^{nd}$ neighbor, etc.) divided by the longest chain length. To construct off-diagonal feature vector, we first sum the atom feature vectors of the connected atoms. This vector is then stacked with one-hot encoded bond type vector, normalized bond length, and one-hot encoded ring size vector to become the off-diagonal feature vector. Since there are no bond features for a focal atom, the bond feature vector channels on-digonal are also filled with 0s. Therefore, **both on and off-diagonal feature vectors have the same dimension**.

There are also two variants of the off-diagonal feature vector. For the G tensor, coordinate and distance channles are excluded. For the GDR tensor, to match the size of the on-diagonal feature vector, every off-diagonal feature vector has three more coordinate channels filled with 0s, and an additional distance channel holding the Euclidean distance between two connected atoms. This off-diagonal feature vector is obtained for all atom pairs, making the proposed tensor fully-connected and symmetric.

A tensor encoding of the benzene ring is illustrated in Fig.1. Having obtained the tensor representation, a naive way of building a generative model is to apply a convolutional neural network directly on the tensor, and train it to predict a distribution over the inter-atomic distances. We utilize a standard UNet (Ronneberger et al., 2015) structure to map the input tensor to a probability distribution over a distance matrix containing all pair-wise Euclidean distances. Distance matrices are then sampled and converted to valid conformations following the same method presented in GraphDG (Simm & Hernández-Lobato, 2020). We refer to this model as the NaiveUNet. More details of the NaiveUNet can be found in Sec.A.4.

This naive model achieves unsatisfactory performance as shown in Tab.1 and Tab.9, merely outperforming GraghDG and is far from that of the state-of-the-art. There are two major issues to this approach. First, with a small kernel size ($3 \times 3$ used in the UNet), it takes many convolution layers to achieve information aggregation between atoms that are far apart; it does not take full advantage of high-order bonds (chemical or virtual) already made available in the input tensor. Secondly, the output size grows quadratically with the number of atoms, as compared to only linear growth in the reconstruction-based or direct generation methods. The solution to the first issue is rather simple, obtained by increasing the kernel size to expand its "field of view". On the other hand, solving the second issue requires elevating the naive two-step generative model to a direct one.

## 2.3 Extended kernel and Attention Mechanism

We observe that every row or column of the proposed tensor contains global information encompassing a focal atom and all of its connected atoms (by both chemical and virtual bond). This motivates our **second main idea** which is to extend the length of the kernel to the length of the tensor graph while keeping the width unaltered. This idea has a profound implication; *global* information from the immediate neighbors, all their connected atoms, and all the bond features can be aggregated onto the focal atom in a single convolution operation. In contrast, achieving the same aggregation may require many layers of propagation for the naive model and other GNN-based models. A direct consequence of this modification is that only 1D convolution is permitted. With multiple kernels being applied simultaneously, each stride of these kernels generates a feature vector for a single atom. An illustration of the 1D convolution operation is shown in Fig.2.

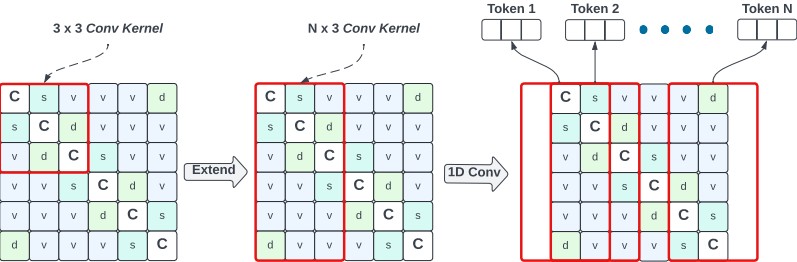

Figure 2: Extending kernel and 1D convolution.

We further observe that the generated feature vectors resemble the token-like feature vectors used in language modeling. This observation combined with the proven success of attention mechanism in other related work leads to the selection of transformer architecture as the backbone of our generative model. A significant advantage of using transformer's self-attention mechanism is, similar to the extended kernel, it enables a global information aggregation from and for all atoms. It also eliminates the need to maintain separated atom and bond features at each step of feature transformation. We present further insight and a more detailed analysis of the adavantage of this input feature engineering in Sec.A.1. There is also an interesting equivalence between the information aggregation achieved by a fully-connected MPNN (Gilmer et al., 2017) and running a $1 \times 1$ convolution operation over the proposed input tensor, as detailed in Sec.A.2.

## 2.4 Putting everything together

**Conditional variational autoencoder framework.** We aim at obtaining a generative model $p_\theta(R|G)$ that approximates the Boltzmann distribution through Maximum Likelihood Estimation. Particularly, given a set of molecular graphs $G$ and their respective ground-truth conformations $R$, we wish to maximize the following objective.

$$\log p_\theta\left(R|G\right) = \log \int p\left(z\right) p_\theta\left(R|z, G\right) dz \tag{1}$$

A molecular graph can have many random conformations. We assume this randomness is driven by a latent random variable $z \sim p\left(z\right)$, where $p\left(z\right)$ is a known distribution e.g. a standard normal distribution. As $p_\theta\left(R|z, G\right)$ is often modeled by a complex function e.g. a deep neural network, evaluation of the integral in Eq.1 is intractable. Instead, we resort to the same techniques proposed in the original VAE (Kingma & Welling, 2014) to establish a tractable lower bound for Eq.1.

$$\log p_\theta\left(R|G\right) \geq \mathbb{E}_{q_w\left(z|R,G\right)}\left[\log p_\theta\left(R|z, G\right)\right] - D_{KL}\left[q_w\left(z|R, G\right) || p\left(z\right)\right] \tag{2}$$

where $D_{KL}$ is the Kullback-Leibler divergence and $q_w\left(z|R, G\right)$ is a variational approximation of the true posterior $p\left(z|R, G\right)$. We assume $p\left(z\right) = \mathcal{N}\left(0, \boldsymbol{I}\right)$ and $q_w\left(z|R, G\right)$ is a diagonal Gaussian distribution whose

means and standard deviations are modeled by a transformer encoder. The input of this transformer encoder is the proposed tensor containing both the coordinate and distance information. We denote this tensor the GDR tensor. On the other hand, $p_\theta(R|z, G)$ is further decomposed into two parts: a decoder $p_{\theta_2}(R|z, \sigma_{\theta_1}(G))$ for predicting conformation directly and another encoder $\sigma_{\theta_1}(G)$ for encoding the 2D molecular graph. The input tensor for $\sigma_{\theta_1}(G)$ is absent of coordinate and distance information, and is therefore denoted the G tensor. Both encoders share the same standard transformer encoder structure. However, there is a minor modification to the transformer structure for the decoder. Specifically, the Query, Key matrices for the first multi-head attention layer are computed based on the output vectors of $\sigma_{\theta_1}(G)$, and the Value matrices come directly from the reparameterization of the output of $q_w(z|R, G)$, as $z = \mu_w + \Sigma_w \epsilon$, where $\mu_w$ and $\Sigma_w$ are the predicted mean and standard deviation respectively. $\epsilon$ is sampled from $\mathcal{N}(0, \boldsymbol{I})$. We present the complete picture of how the two encoders and the decoder are arranged in a CVAE framework in Fig.3.

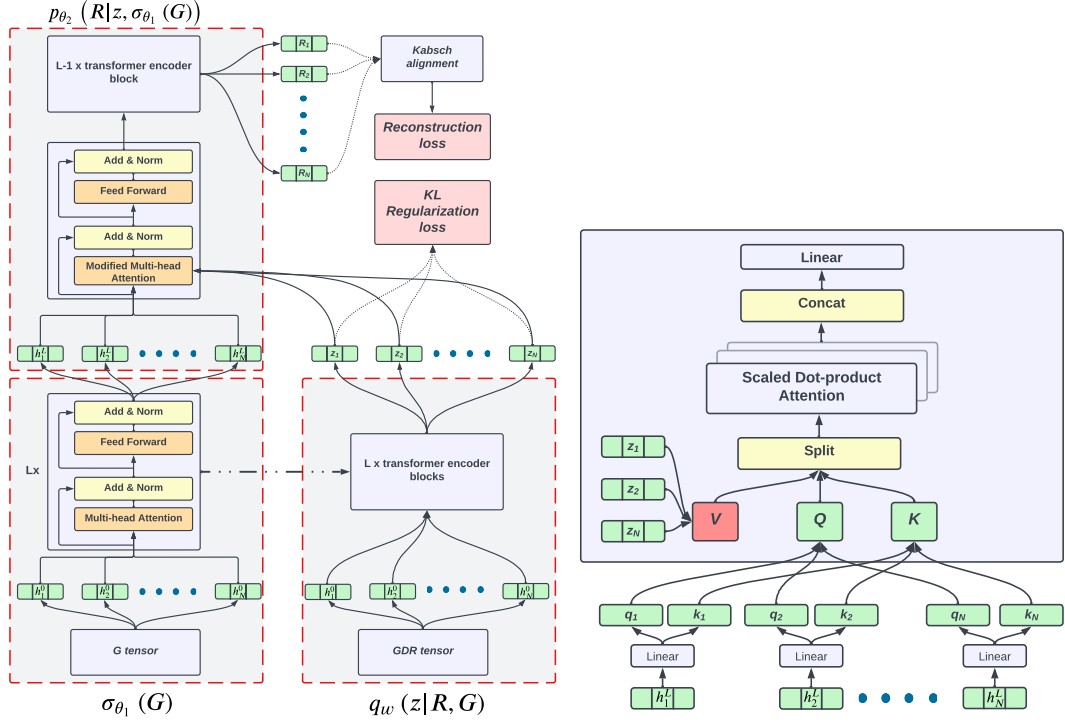

Figure 3: Variational AutoEncoder framework (left) and modified multi-head attention (right)

**Intuition behind the modified attention.** There are multiple ways to join together the output of the two encoders to form the input to the final decoder. Popular methods include stacking or addition. We tried both these methods with unsatisfactory performance. We notice that, due to direct stacking or addition of the sampled output of $q_w$ onto the output of $\sigma_{\theta_1}$, attention weights computed in the first layer of the decoder are easily overwhelmed by random noise of the sampled values, and become almost indiscernible[1]. This leads to ineffective information aggregation which is then further cascaded through the remaining attention layers. Intuitively, in the first attention layer, the attention weights dictating how much influence an atom exerts on the other should predominantly be determined by the graph structure, and remain stable for the same molecule. Further, attention weights are computed by Query and Key matrices. Therefore, these two matrices should stay stable for the same graph. This motivates **our third and final main idea**; that is, we compute Query and Key matrices only from the output $\{h_1^L, ..., h_N^L\}$ of $\sigma_{\theta_1}$, and attribute the variation in conformation to the Value matrices which are directly sampled from $\{z_1, ..., z_N\} \sim q_w$. The resultant information aggregation is much more meaningful and each output vector corresponding to an individual atom carries distinct features, facilitating information aggregation of the ensuing attention layers.

---

[1]Imagine a mixture model with randomly varying mixture weights.

**Learning to achieve approximate Roto-translation invariant loss.** Following ConfVAE (Xu et al., 2021b), we formulate the reconstruction loss as.

$$- \log p_\theta \left( R | z, G \right) = - \sum_{i=1}^{N} \sum_{j=1}^{3} \left( R_{ij} - A \left( \hat{R}, R \right)_{ij} \right)^2 \tag{3}$$

where $A \left( \cdot \right)$ is a function aligning the predicted conformation $\hat{R}$ onto the reference conformation $R$. We choose Kabsch algorithm (Arun et al., 1987) as the alignment method which translates and rotates the predicted conformation onto its corresponding ground-truth before loss computation. This makes the reconstruction loss roto-translation invariant.

Simultaneously, minimizing the KL-loss component $D_{KL} \left[ q_w \left( z | R, G \right) || p \left( z \right) \right]$ compels the output of the posterior encoder to adhere to a standard normal distribution. Despite this minimization promoting convergence, achieving exact equality $q_w \left( z | R, G \right) = p \left( z \right)$ in practice is challenging, especially in the presence of SE(3) transformations of the input $R$. Consequently, upon convergence, the objective function defined in Eq.2 only achieves approximate roto-translation invariance.

**Direct conformation generation at inference time.** To generate a single conformation, we first construct the G tensor of a molecular graph and obtain a single latent sample $\{z_1, ...z_N\}$ from a standard diagonal Gaussian distribution. The G tensor is passed through $\sigma_{\theta_1}$ encoder to produce $\left\{ h_1^L, ..., h_N^L \right\}$ which is then combined with the latent sample via the modified multi-head attention mechanism. The output of this modified attention layer further goes through $L - 1$ standard attention layers to be transformed to the final conformation. **The entire generation process depends only on a 2D molecular graph, and requires a single sampling step and a single pass of the TensorVAE model**.

## 3 Experiment

In this section, we first elaborate on the implementation details of the TensorVAE model including determining the size of the input tensors, network architecture and how the entire framework is trained end-to-end. We then present conformation generation experiment results of the proposed TensorVAE on three benchmark data-sets, including GEOM-QM9, GEOM-Drugs and Platinum data-sets. While the GEOM datasets contain unbound conformations of molecules, the Planinum dataset contains molecular conformations bound to their respective protein targets. The generative performance of the proposed model is compared to those of **15** state-of-the-art baselines. In addition to conformation generation, in Sec.A.8, we further demonstrate the effectiveness of information aggregation of the proposed TensorVAE architecture by briefly comparing the molecular property prediction performance of the proposed method against **7** more state-of-the-art baselines on the MolecularNet (Wu et al., 2018) benchmark.

### 3.1 Experiment setup

**Dataset.** Following existing work (Luo et al., 2021; Shi et al., 2021; Xu et al., 2021b;a; 2022; Zhou et al., 2023), we utilize the GEOM data-set for evaluating the performance of the proposed TensorVAE. GEOM contains 37 million energy and statistical weight annotated molecular conformations corresponding to 450,000 molecules (Axelrod & Gómez-Bombarelli, 2022). This dataset is further divided into two constituent datasets, Drugs and QM9. The Drugs dataset covers 317,000 median-sized molecules averaging 44.4 number of atoms. The QM9 dataset contains 133,000 smaller molecules averaging only 18 atoms.

We follow Xu et al. (2022) to randomly select 40,000 molecules from each dataset to form the training set. For each molecule, we choose the top 5 most likely[2] conformations. This results in 200,000 training conformations for each train set. For validation set, we randomly sample 2,500 conformations for both Drugs and QM9 experiments. Finally, for testing, following (Shi et al., 2021; Xu et al., 2022), we randomly select

---

[2]Ranked by their Boltzmann weight.

200 molecules each with more than 50 and less than 500 annotated conformations from QM9, and another 200 with more than 50 and less than 100 annotated conformations from Drugs[3].

The GEOM dataset contains conformations of molecules that are not bound to any specific target. To assess the proposed model's ability to generate ligand-protein bound conformations, we additionally evaluate its performance using the Platinum dataset (Friedrich et al., 2017). The Platinum dataset is derived from the Pretein Data Bank (Berman et al., 2000) and consists of two high-quality ligand-protein bound conformation dataset: a comprehensive dataset and a diversified subset of 4,626 and 2,912 structures, respectively. Following the setup in (Friedrich et al., 2017), we test the performance of the proposed TensorVAE on the diversified subset.

**Determining input tensor size and atom ordering.** We conduct a basic data analysis on the entire Drugs dataset to determine the $98.5^{th}$ percentile of the number of atoms to be 69, and the percentage of molecules having more than 69 atoms and with more than 50 but less than 100 conformations is only 0.19%. Accordingly, we set the size of the input tensor to $69 \times 69$ for Drugs experiment. On the other hand, we use the maximum number of atoms 30 for QM9 experiment. The channel features for the input tensor include atom types, atom charge, atom chirality, bond type, bond stereo-chemistry and bond in-ring size. For the GDR tensor, we also include 3D coordinate channels and a distance channel. The resulting channel depth is 50 for GDR tensor and 46 for G tensor. The detailed information of these features and their encoding method is listed in Sec.A.5. The ordering of the atoms along the diagonal of the tensor is determined by a **random Depth-First Traversal (DFT)** of the molecular graph.

**Implementation details.** We implement the proposed TensorVAE using Tensorflow 2.5.0. All three transformer encoders of TensorVAE follow the standard Tensorflow implementation in `https://www.tensorflow.org/text/tutorials/transformer`. All of them have 4 layers, 8 heads and a latent dimension of 256. Both QM9 and Drugs experiments share the same network architecture and hyper-parameter configuration. We present the detailed training hyperparameter configuration in Sec.A.3.

**Evaluation metrics.** We adopt the widely accepted coverage score (COV) and matching score (MAT) (Shi et al., 2021) to evaluate the performance of the proposed TensorVAE model. These two scores are computed as;

$$\text{COV}\left(\mathbb{C}_g, \mathbb{C}_r\right) = \frac{1}{|\mathbb{C}_r|} \left| \left\{ R \in \mathbb{C}_r | \text{RMSD}\left(R, \hat{R}\right) \leq \delta, \forall \hat{R} \in \mathbb{C}_g \right\} \right| \tag{4}$$

$$\text{MAT}\left(\mathbb{C}_g, \mathbb{C}_r\right) = \frac{1}{|\mathbb{C}_r|} \sum_{R \in \mathbb{C}_r} \min \text{RMSD}\left(R, \hat{R}\right) \tag{5}$$

where $\mathbb{C}_g$ is the set of generated conformations and $\mathbb{C}_r$ is the corresponding reference set. The size of $\mathbb{C}_g$ is twice of that of $\mathbb{C}_r$, as for every molecule, we follow (Xu et al., 2022) to generate twice the number of conformations as that of reference conformations. $\delta$ is a predefined threshold and is set to 0.5Å for QM9 and 1.25Å for Drugs respectively (Shi et al., 2021) . RMSD stands for the root-mean-square deviation between $R$ and $\hat{R}$, and is computed using the `GetBestRMS` method in the `RDKit` (Riniker & Landrum, 2015) package. While COV score measures the ability of a model in generating diverse conformations to cover all reference conformations, MAT score measures how well the generated conformations match the ground-truth. A good generative model should have a high COV score and a low MAT score.

To evaluate the accuracy of the proposed model on the Platinum dataset, we employ two metrics: the root-mean-square deviation (RMSD) for four ensemble sizes (10, 50, 250, and 500) and the percentage of molecules with RMSD within specified thresholds (0.5, 1.0, 1.5, and 2) for two ensemble sizes (50 and 250). In terms of generative speed evaluation, we calculate and compare the mean and median generation times for the four ensemble sizes across all 2,912 molecules.

**Baselines**. We first compare the generative performance of the proposed TensorVAE model to those of 1 classical `RDKit` method; 5 distance-based methods including GraphDG, CGCF, ConfVAE, ConfGF and DGSM; 2 reconstruction-based methods including GeoDiff and Uni-Mol; 3 direct methods including CVGAE,

---

[3]This limit on the number of conformations for testing molecules is taken directly from `https://github.com/DeepGraphLearning/ConfGF` which is also followed by all other compared methods in the GEOM experiment.

GeoMol, and DMCG. For the Platinum dataset, we also incorporate 4 classical methods, namely Ballon DG and Ballon GA (Vainio & Johnson, 2007), MultiConf-Dock (Sauton et al., 2008) and ETKDG (Riniker & Landrum, 2015).

We then compare the molecular property prediction performance of the proposed model (specifically the GDR encoder) to 7 more strong baselines comprising D-MPNN (Yang et al., 2019), AttentiveFP (Xiong et al., 2019), N-Gram (Liu et al., 2019), PretrainingGNN (Hu et al., 2020), GROVER (Rong et al., 2020), GEM (Fang et al., 2022) and finally again Uni-Mol.

### 3.2 Results and Discussion

**Unbound conformation generation**. The COV and MAT scores for all compared methods on both QM9 and Drugs datasets are presented in Tab.1. The proposed TensorVAE achieves the state-of-the-art generative performance. Additionally, **we have conducted 4 ablation** studies on the input feature engineering method in Sec.3.3 to demonstrate why 1D convolution with a $N \times 3$ kernel is crucial to achieving a good generative performance. While none of the cited baselines quantify confidence of their results, we have included standard deviations in all our results.

In Tab.1, TensorVAE[REF] results are obtained by running test on the same set of test data[4] that is adopted by all other baselines. While TensorVAE employs the same test set for evaluation, it's essential to note that the training dataset differs from that in ConfGF. Nevertheless, we have conducted a thorough examination to confirm that none of the molecules in the ConfGF test set are included in our training dataset.

Additionally, we observed that the **ConfGF DRUGS test set has a maximum of 71 heavy atoms per molecule**, exceeding our predetermined maximum of 69 atoms by 2. While there are only 3 molecules with more than 69 heavy atoms, we do not anticipate a significant performance change by allowing TensorVAE to handle an additional 2 atoms. Therefore, we opt not to retrain TensorVAE for this test set. To ensure a fair comparison, for these 3 molecules, we assume a worst-case scenario where the trained TensorVAE can only achieve a MAT score of 2Å and a COV score of 0%. The TensorVAE[REF]'s Mean/Median MAT and COV scores for the DRUGS dataset are computed under this worst-case scenario. For the QM9 dataset, as we have already used the maximum number of heavy atoms, TensorVAE[REF]'s results are obtained as usual.

On the other hand, TensorVAE[1] results have been obtained on a set of random testset, selected based on the same filtering condition proposed in ConfGF and having a maximum number of heavy atoms of 69 per molecule. This set of 200 molecules contains 23,079 and 14,396 testing conformation for QM9 and Drugs, respectively. TensorVAE[1] results and standard deviations are obtained by running 10 experiements each with a different random seed on the same 200 testing molecules.

TensorVAE[2] results are obtained by running 10 experiements each with a different random seed as well as a different set of 200 testing molecules. In this setting, both testsets contain 2,000 testing molecules, amounting to more than $280k$ and $140k$ testing conformations for QM9 and Drugs, respectively. **The number of testing conformations is more than** $70\%$ **of that of training conformations**. Attaining consistent performance on this much larger testset consolidates the generalization capability of the proposed TensorVAE, and **verifies its robustness under random permutation of atom ordering**. Additionally, as noted by Xu et al. (2022), Eqs.4 and 5 are only the *recall* scores. We also present the *precision* scores results in Tab.11 of Sec.A.6, where TensorVAE again achieves the state-of-the-art performance with a considerable margin.

Xu et al. (2021b) discovered that the quality of conformations generated by deep generative models can be further refined by an additional empirical force field (FF) (Halgren, 1996) optimization procedure. Uni-Mol also leverages FF optimization to improve its generative performance. Different from GeoDiff which reconstructs a valid conformation directly from random noisy coordinates, Uni-Mol simply refines an initial conformation optimized by `RDKit` (using ETKGD with FF (Riniker & Landrum, 2015)).

---

[4]This dataset is available for download at `https://github.com/DeepGraphLearning/ConfGF`. Originally generated in ConfGF, it serves as the common dataset across all compared baselines. For the 200 testing molecules, the total numbers of annotated conformations are 22,408 and 14,324 for QM9 and Drugs, respectively. As all compared baselines utilize the same test set, including standard deviation as an uncertainty measure is unnecessary.

Table 1: Performance comparison between TensorVAE and 10 baselines on GEOM dataset.

| Models | QM9 | | | | Drugs | | | |
|---|---|---|---|---|---|---|---|---|
| | COV (%) ↑ | | MAT (Å) ↓ | | COV (%) ↑ | | MAT (Å) ↓ | |
| | Mean | Median | Mean | Median | Mean | Median | Mean | Median |
| RDkit | 83.26 | 90.78 | 0.3447 | 0.2935 | 60.91 | 65.70 | 1.2026 | 1.1252 |
| CVGAE | 0.09 | 0.00 | 1.6713 | 1.6088 | 0.00 | 0.00 | 3.0702 | 2.9937 |
| GraphDG | 73.33 | 84.21 | 0.4245 | 0.3973 | 8.27 | 0.00 | 1.9722 | 1.9845 |
| CGCF | 78.05 | 82.48 | 0.4219 | 0.3900 | 53.96 | 57.06 | 1.2487 | 1.2247 |
| ConfVAE | 80.42 | 85.31 | 0.4066 | 0.3891 | 53.14 | 53.98 | 1.2392 | 1.2447 |
| ConfGF | 88.49 | 94.13 | 0.2673 | 0.2685 | 62.15 | 70.93 | 1.1629 | 1.1596 |
| GeoMol | 71.26 | 72.00 | 0.3731 | 0.3731 | 67.16 | 71.71 | 1.0875 | 1.0586 |
| DGSM | 91.49 | 95.92 | 0.2139 | 0.2137 | 78.73 | 94.39 | 1.0154 | 0.9980 |
| GeoDiff | 92.65 | 95.75 | 0.2016 | 0.2006 | 88.45 | 97.09 | 0.8651 | 0.8598 |
| DMCG | 94.98 | 98.47 | 0.2365 | 0.2312 | 91.27 | 100 | 0.8287 | 0.7908 |
| TensorVAE[REF] | 97.79 | 100 | 0.1985 | 0.1951 | 93.05 | 98.98 | 0.8087 | 0.7866 |
| TensorVAE[1] | **98.11** ±0.25 | **100** ±0 | **0.1970** ±0.0016 | **0.1926** ±0.0027 | **94.91** ±0.35 | **100** ±0 | **0.7789** ±0.0027 | **0.7585** ±0.0076 |
| TensorVAE[2] | 97.11 ±0.31 | 100 ±0 | 0.2041 ±0.0046 | 0.1920 ±0.007 | 93.34 ±1.17 | 99.90 ±0.31 | 0.8074 ±0.0135 | 0.7927 ±0.0186 |

*Bold font indicates best result. Results for RdKit, CVGAE, GraphDG, CGCF, ConfGF are taken from (Shi et al., 2021); all other results are taken from (Zhou et al., 2023). Values following ±are standard deviations.

For a fair comparison, we exclude deep generative models relying on FF optimization from Tab.1 and compare their performances separately in Tab.2. Again, the proposed TensorVAE with FF optimization outperforms all of them with a significant margin.

Table 2: Performance comparison between methods **with FF optimization** on GEOM Drugs dataset

| Method | COV(%) ↑ | | MAT(Å) ↓ | |
|---|---|---|---|---|
| | Mean | Median | Mean | Median |
| CVGAE | 83.08 | 95.21 | 0.9829 | 0.9177 |
| GraphDG | 84.68 | 93.94 | 0.9129 | 0.9090 |
| Uni-Mol | 91.91 | 100 | 0.7863 | 0.7794 |
| CGCF | 92.28 | 98.15 | 0.7740 | 0.7338 |
| ConfVAE | 91.88 | 100 | 0.7634 | 0.7312 |
| GeoDiff | 92.27 | 100 | 0.7618 | 0.7340 |
| TensorVAE[REF] | 93.36 | 98.18 | 0.7267 | 0.7032 |
| TensorVAE[2] | **94.74** ±0.66 | **100** ±0 | **0.6985** ±0.012 | **0.6845** ±0.0196 |

*Results for CVGAE, GraghDG, CGCF, and ConfVAE are taken from (Xu et al., 2021b); GeoDiff and Uni-Mol results are from their source paper.

In terms of simplicity, the proposed TensorVAE uses a standard transformer encoder and a simple Kabsch alignment loss. On the other hand, due to the lack of effective input feature engineering, both DMCG and Uni-Mol require design of sophisticated network architectures and complex loss functions to achieve a good generative performance. A direct consequence of these complicated designs is a large number of model parameters, as shown in Tab.3.

Table 3: Comparison of number of parameters among TensorVAE, DMCG and Uni-Mol.

| Method | Number of parameters |
| --- | --- |
| DMCG | 128M |
| Uni-Mol | 47.81M |
| TensorVAE training | 11.5M |
| TensorVAE inference | 6.65M |

*TensorVAE has less number of parameters during inference time as the GDR encoder is not needed.

In terms of efficiency, TensorVAE is a direct generative model capable of producing conformation from a 2D molecular graph in a single step. It takes only 62 seconds using a single Xeon 8163 CPU to decode 200 QM9 molecules, and 128 seconds for 200 Drug molecules. In comparison, GeoDiff requires $5,000$ diffusion steps per conformation, and takes around $8,500$ seconds for decoding 200 QM9 molecules and $11,500$ seconds for decoding 200 Drugs molecules on a single Tesla V100 GPU. **The proposed TensorVAE achieves more than** $100\times$ **speed up**. Finally, some samples of the TensorVAE generated conformations are shown in Fig.4.

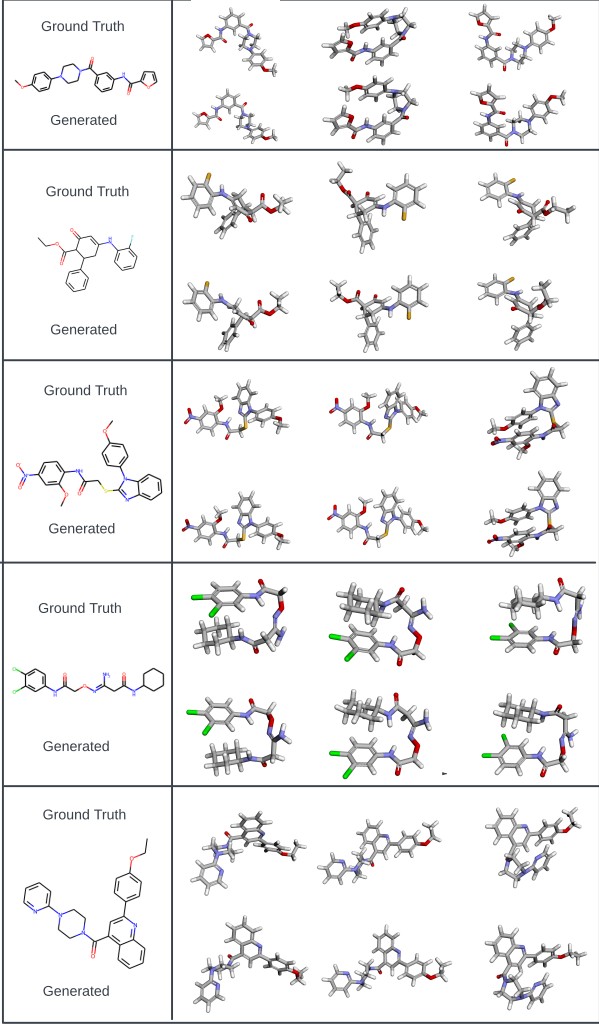

Figure 4: Generated samples by the TensorVAE

**Protein-ligand bound conformation generation.** The performance evaluation of the proposed model on generating ligand-bound conformation is vital to establish its potential application in high throughput virtual screening of drug candidates. Following the setup in Friedrich et al. (2017), we further compare the performance of the proposed TensorVAE model with 5 popular baselines on the Platinum dataset. We took the TensorVAE model trained on the GEOM-drugs conformations, and applied it directly to the **Platinum diverse dataset** for conformer ensemble generation. Before presenting the evaluation results on the Platinum dataset, we would like to first emphasize the difference between the Platinum dataset which was proposed in Friedrich et al. (2017) and the GEOM dataset which we have used to train TensorVAE. While the GEOM-drugs dataset mainly contains vacuum conformer-rotamer ensembles that are generated using semi-empirical density functional theory, the Platinum dataset only includes protein-bound ligand conformations. The energy states of conformers bound to a protein target are different from those of the stable unbound conformers. The underlying distributions governing the generation of these two datasets also differ significantly. Testing on the Platinum dataset without any retraining or finetuning creates a **distribution shift** from that of the GEOM training data. Inevitably, this will lead to performance degradation of the proposed TensorVAE. However, evaluating the proposed TensorVAE on the Platinum dataset remains valuable for assessing its ability to generalize and accurately generate valid ligand-protein bound conformations, despite being trained solely on unbound conformations.

We have repeated the experiments from Table 3 to Table 6 in Friedrich et al. (2017). The results of these experiments are presented below.

Table 4: Arithmetic Mean and Median RMSD in Å Obtained for the Platinum Diverse Dataset.

| Maximum ensemble size | 10 | | 50 | | 250 | | 500 | |
|---|---|---|---|---|---|---|---|---|
| | Mean | Median | Mean | Median | Mean | Median | Mean | Median |
| Balloon DG | 1.10 | 0.97 | 1.00 | 0.86 | 0.92 | 0.77 | 0.89 | 0.74 |
| Balloon GA | 1.22 | 1.10 | 0.90 | 0.80 | 0.72 | 0.63 | 0.67 | 0.58 |
| RDKit | 1.00 | 0.89 | 0.77 | 0.64 | 0.63 | 0.52 | 0.59 | 0.48 |
| ETKDG | 0.98 | 0.87 | 0.77 | 0.66 | 0.63 | 0.54 | 0.59 | 0.51 |
| Multiconf-DOCK | 0.99 | 0.89 | 0.84 | 0.72 | 0.80 | 0.69 | 0.80 | 0.69 |
| TensorVAE | 1.02 | 0.95 | 0.85 | 0.77 | 0.73 | 0.67 | 0.69 | 0.63 |

Table 5: Fraction of Structures of the Platinum Diverse Dataset Successfully Reproduced within a Specified RMSD Threshold.

| Maximum ensemble size | 50 | | | | 250 | | | |
|---|---|---|---|---|---|---|---|---|
| Minimum accuracy [Å] | 0.5 | 1.0 | 1.5 | 2.0 | 0.5 | 1.0 | 1.5 | 2.0 |
| Balloon DG | 0.29 | 0.57 | 0.77 | 0.92 | 0.33 | 0.62 | 0.81 | 0.92 |
| Balloon GA | 0.30 | 0.72 | 0.90 | 0.97 | 0.43 | 0.84 | 0.96 | 0.99 |
| RDKit | 0.39 | 0.71 | 0.89 | 0.96 | 0.48 | 0.82 | 0.95 | 0.98 |
| ETKDG | 0.36 | 0.72 | 0.91 | 0.97 | 0.45 | 0.83 | 0.95 | 0.99 |
| Multiconf-DOCK | 0.32 | 0.68 | 0.87 | 0.96 | 0.34 | 0.71 | 0.89 | 0.97 |
| TensorVAE | 0.27 | 0.65 | 0.89 | 0.97 | 0.34 | 0.76 | 0.95 | 0.99 |

Although the proposed TensorVAE is trained solely on unbound conformations, it demonstrates comparable performance to 5 popular baselines in terms of accurately generating ligand-protein bound conformations (Tab.4 and Tab.5), which serves to validate its generalization capability. More specifically, it demonstrates a slight performance advantage over Balloon DG/GA and multiconf-Dock; however, it falls short of matching the performance achieved by RDkit and ETKDG. This result appears to contradict the findings obtained from the GEOM dataset, where the proposed TensorVAE outperformed RDkit.

The main reason of this contradiction could be attributed to the **distribution shift** or **dataset shift** between training and testing. Additionally, for constructing the training dataset, we sampled 40,000 molecules

from GEOM drugs dataset and only retained the top-5 conformations with the highest Boltzmann weight for each molecule. These conditions further restrict the energy search space for conformation generation. Consequently, the Boltzmann distribution approximated (and learned) by the proposed TensorVAE might not be directly suited to prediction of ligand-bound conformations without further fine-tuning.

Table 6: Arithmetic Mean and Median Ensemble Sizes Measured for the Platinum Diverse Dataset.

| Maximum ensemble size | 10 | | 50 | | 250 | | 500 | |
|---|---|---|---|---|---|---|---|---|
| | Mean | Median | Mean | Median | Mean | Median | Mean | Median |
| Balloon DG | 10 | 10 | 50 | 50 | 249 | 250 | 498 | 500 |
| Balloon GA | 9 | 10 | 49 | 50 | 244 | 250 | 487 | 500 |
| RDKit | 10 | 10 | 50 | 50 | 250 | 250 | 500 | 500 |
| ETKDG | 10 | 10 | 50 | 50 | 250 | 250 | 500 | 500 |
| Multiconf-DOCK | 9 | 10 | 36 | 50 | 78 | 57 | 80 | 57 |
| TensorVAE | 10 | 10 | 50 | 50 | 250 | 250 | 500 | 500 |

In terms of generative capability (Tab.6), the proposed TensorVAE is able to generate the complete 10-, 50-, 250-, and 500-conformers ensemble sizes for all molecules which puts it head-to-head against RDKit and ETKDG. In terms of generative speed (Tab.7), as the proposed TensorVAE only needs a single pass of the neural network to generate conformations for each ensemble size, its mean and median runtimes (measured on a single core of Xeon 8163 CPU) are significantly faster than the other compared methods.

Table 7: Arithmetic Mean and Median Runtimes in Seconds Measured for the Platinum Diverse Dataset.

| Maximum ensemble size | 10 | | 50 | | 250 | | 500 | |
|---|---|---|---|---|---|---|---|---|
| | Mean | Median | Mean | Median | Mean | Median | Mean | Median |
| Balloon DG | 6 | 5 | 27 | 24 | 132 | 117 | 260 | 260 |
| Balloon GA | 4 | 3 | 19 | 17 | 105 | 98 | 256 | 234 |
| RDKit | 1 | 1 | 5 | 4 | 22 | 18 | 42 | 34 |
| ETKDG | 1 | 1 | 4 | 3 | 16 | 12 | 32 | 23 |
| Multiconf-DOCK | 5 | 1 | 8 | 2 | 15 | 3 | 15 | 3 |
| TensorVAE | <1 | <1 | <1 | <1 | 1 | 1 | 2 | 2 |

### 3.3 Ablation studies

In this section, we further demonstrate the effectiveness and necessity of running an 1D convolution with $N \times 3$ kernels over the proposed input tensor through 4 ablation studies on GEOM drugs dataset. We also show that the transformer attention mechanism is also an important contributing factor for a competitive generative performance.

**Why is 1D convolution necessary**. We have shown a model based on a $3 \times 3$ kernel in Sec.A.4 called NaiveUNet. Here, we provide a more detailed analysis of why NaiveUNet produces unsatisfactory result. The primary reason for this poor performance is the "field of view" of a conventional $d \times d$ ($d < N$) kernel only sees a partial connection pattern of a focal atom. In comparison, a $N \times 3$ kernel's "field of view" encompasses the complete connection pattern of a focal atom. We further observe that when applying a $3 \times 3$ kernel filter to the top left region of the proposed tensor, its field of view only includes a focal atom, its two neighboring atoms and how the focal atom is connected to them. There are two main disadvantages associated with this. Firstly, it only achieves a 1-hop information aggregation. Secondly when the $3 \times 3$ kernel moves to an off-diagonal part of the tensor, where most connections are virtual bonds (as atoms of a molecule are often sparsely connected), information aggregation occurs mostly between atoms that are not chemically connected and is therefore less meaningful than that on the diagonal part of the tensor. For these two reasons, the NaiveUNet's performance on the GEOM Drugs dataset is the worst as shown in Tab.9.

**What happens if we remove all virtual bonds**. Notice that if we remove all the virtual bonds in each column and still run a $N \times 3$ kernel through the tensor, its "field of view" is a "2-hop atomic-environment" (because the focal atom can "see" how neighboring atoms are chemically connected to all their direct neighbors). Another observation is that after removing all virtual bonds, each column does not correspond to a fully-connected MPNN. Therefore it no longer enables a global information aggregation. The conformation generation results of this variant of TensorVAE on Drugs dataset is shown as as TensorVAE abla1 in table below. It is observed that due to a less effective local information aggregation as a result of removing all virtual bonds (and related atom features), the performance is worse than of the complete TensorVAE version.

**What happens if a $N \times 1$ kernel is used**. The third ablation study concerns with using a $N \times 1$ kernel with a smaller "field of view" as compared to that of a $N \times 3$ kernel. Its performance on Drugs dataset is shown as TensorVAE abla2 in Tab.9. It performs slightly better than the ablation removing all virtual bonds. The reason is that though its field of view is smaller, it still achieves a global information aggregation for the focal atom. Nevertheless, it underperforms the complete TensorVAE version due to a smaller "field of view" for information aggregation.

**What happens if a $1 \times 1$ kernel is used**. This setup corresponds to connecting a fully-connected MPNN (GNN) with a standard transformer backbone for conformation generation. Since using a $1 \times 1$ kernel leads to a model with a signicantly less model compacity as compared to the models in previous ablation studies, we experimented with 6 hyper-parameter configurations listed as following to ensure this variant has roughly the same model capacity (number of parameters).

Table 8: Experimental setups for $1 \times 1$ kernel.

| Model name | Embedding size | KL weight schedule | No. of transformer layers | No. of parameters |
|---|---|---|---|---|
| GNN_base | 256 | same as TensorVAE | 4 | 6.5M |
| GNN_large1 | 320 | same as TensorVAE | 4 | 11M |
| GNN_large2 | 256 | same as TensorVAE | 6 | 10M |
| GNN_large3 | 320 | 1e-5 doubling every 16 epochs | 6 | 11M |
| GNN_large4 | 320 | 1e-6 doubling every 16 epochs | 6 | 11M |
| GNN_large5 | 320 | 1e-7 doubling every 16 epochs | 6 | 11M |

There are two ways to increase the number of parameters of the MPNN-based variant to match that of the TensorVAE employing a $N \times 3$ kernel for a fair comparison, including a larger embedding size and more transformer layers. These two setups correspond to large 1 and large 2. Unfortunately, training for these 3 setups failed to reduce RMSD error after more than 10 epochs of training; we kept facing the KL vanishing problem. To tackle this, we experimented with 3 more configurations (large 3,4 and 5) with much lower KL weights and shorter step period to force training to focus more on reducing the RMSD loss. Unfortunately again, after more than 40 epochs (25+ hours) of training all three efforts have also failed to resolve this issue. We have included the training and validation curve for all 6 experiments in Fig.5.

It is observed that for all cases, while KL error quickly decreases to close to zero, the RMSD loss stays almost constant at 4.0, indicating model's inability to learn. It seems that the MPNN-based models struggle to learn any meaningful information that contribute to producing valid conformations. Instead, they always resort to reducing KL loss which is a much easier learning task. **This fact combined with previous 3 ablation studies manifest an emerging trend that the TensorVAE model's capacity to learn difficult conformation generation task improves with the increase of expressive power of its aggregation mechanism**. In other words, the extra flexibility introduced by the increased kernel size (from $1 \times 1$ to $N \times 3$) is a main contributing factor to the promising performance of the TensorVAE model. Therefore, we conclude that the design choice made to use a $N \times 3$ kernel is sensible and fully justified.

**What happens if the transformer architecture is replaced by a MLP**. In this setup, we replace the transformer encoder block with a MLP block as following:

$$m_i^l = W_2(\mathbf{RELU}(W_1 h_i^l + b_1)) + b_2$$
$$h_i^{l+1} = \mathbf{DropOut}(\mathbf{LayerNorm}(m_i^l))$$

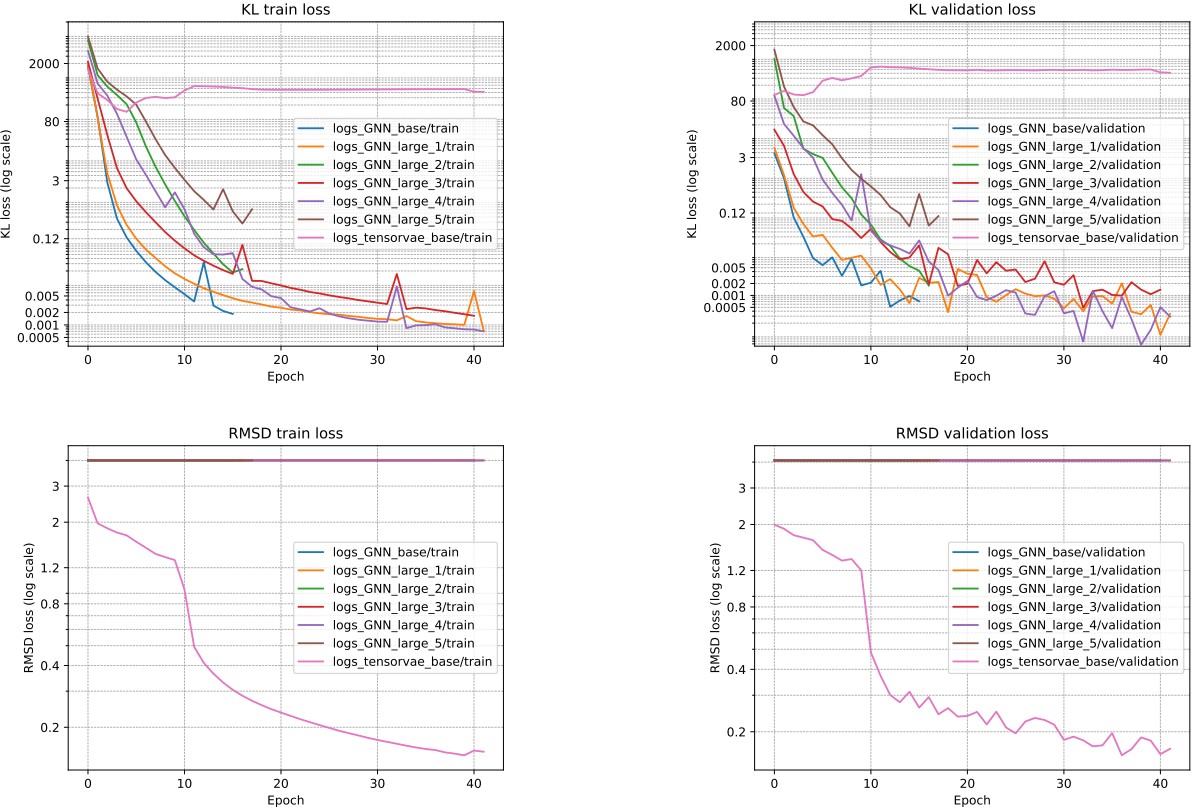

Figure 5: Ablation study: performance comparison between using $N \times 3$ kernel and $1 \times 1$ kernel. The model architectural difference between GNN_large models and the GNN_base model can be found in Tab.8

Where $h_i^l \in R^{256 \times 1}$ is the $l^{th}$ layer output for the $i^{th}$ atom, $W_1 \in R^{1024 \times 256}$, $b_1 \in R^{1024 \times 1}$, $W_2 \in R^{256 \times 1024}$ and $b_2 \in R^{256 \times 1}$. Additionally, due to the absence of attention mechanism, the output $\{h_1^L, ..., h_N^L\}$ of the graph encoders $\sigma_{\theta_1}(G)$ and the sampled latent output $\{z_1, ..., z_N\}$ of the posterior encoder $q_w(z|R, G)$ are simply summed to become the input of the of the decoder $p_{\theta_2}(R|z, \sigma_{\theta_1}(G))$ to generate conformation directly. Akin to the TensorVAE, all three components consist of 4 MLP blocks. The total number of parameters corresponding to this setup is 11M which is similar to that of the TensorVAE.

The dropout rate was initially set to 0.1. In this configuration, we trained the model for 100 epochs and observed a severe overfitting issue, as illustrated in Fig.6, where the training and validation curves of the MLP variant and TensorVAE are compared. The MLP variant exhibited not only a significantly higher KL loss but also a much higher RMSD validation loss compared to both its training loss and that of TensorVAE. Upon observing this behavior, we decided to experiment with higher dropout rates, including 0.3, 0.5, and 0.7, to mitigate overfitting.

After training the model for 100 epochs for each dropout rate, we found that the dropout rate of 0.3 achieved the best validation KL and RMSD losses without encountering any overfitting issues. However, with an increase in the dropout rate, the RMSD training loss decreased (while the validation RMSD error remained the same), and the KL loss increased significantly. This behavior mirrored that of the $1 \times 1$ convolution kernel (fully-connected MPNN) variant mentioned earlier. Essentially, the model increasingly relied on posterior encoder information to reconstruct the conformation and reduce the RMSD error, which is an easier task compared to reconstructing conformation from a 2D molecular graph in the absence of any coordinate information. This trend suggested that a higher dropout rate led to a reduction in the model's capacity to learn.

Despite achieving the best performance among all tested dropout rates, after 120 epochs of training, the MLP variant with 0.3 dropout still performed significantly worse than the TensorVAE with a transformer backbone. Although its RMSD validation loss matched that of the TensorVAE, its KL validation loss was more than double that of the TensorVAE, indicating significantly lower learning capacity. Observing this behavior led us to the conclusion that it was no longer necessary to complete the training to demonstrate the necessity of TensorVAE with a transformer architecture to obtain competitive performance. This experiment suggests that the attention mechanism is also a crucial contributing factor for effective information aggregation among atoms.

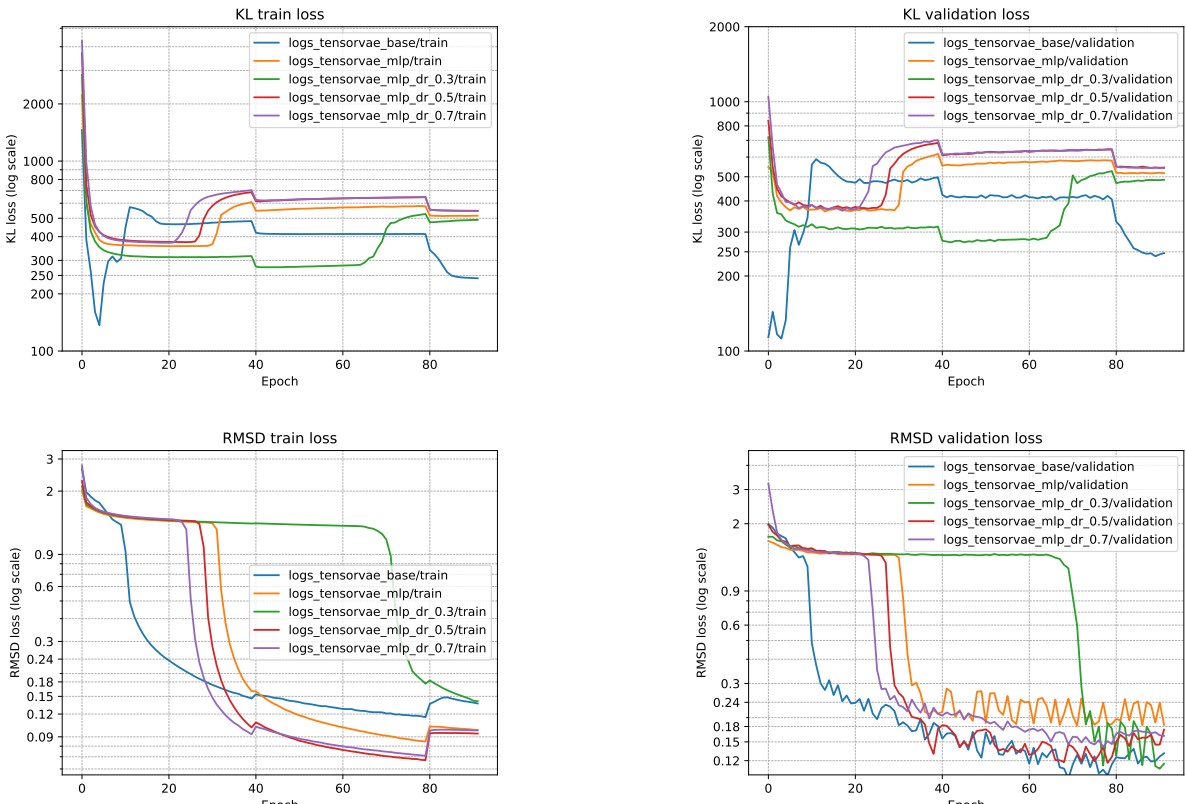

Figure 6: Ablation study: performance comparison among MLP backbones with different dropout rates. While the base TensorVAE with a transformer backbone and the base MLP ablation studay have a default dropout rate of 0.1, the other ablation studies have dropout rates ranging from dr_0.3 to dr_0.7, respectively.

Table 9: Performance comparison among models with different input feature engineering setup on GEOM Drugs dataset

| Method | COV | | MAT | |
|---|---|---|---|---|
| | Mean | Median | Mean | Median |
| NaiveUNet | $52.14 \pm 1.48$ | $51.69 \pm 1.17$ | $1.4322 \pm 0.0247$ | $1.3861 \pm 0.0173$ |
| TensoVAE abla1 | $90.72 \pm 1.54$ | $99.53 \pm 0.64$ | $0.8748 \pm 0.0161$ | $0.8619 \pm 0.0214$ |
| TensoVAE abla2 | $91.04 \pm 1.21$ | $99.74 \pm 0.42$ | $0.8706 \pm 0.0131$ | $0.8561 \pm 0.0204$ |
| TensorVAE | $\mathbf{93.34} \pm 0.35$ | $\mathbf{99.90} \pm 0.31$ | $\mathbf{0.8074} \pm 0.0135$ | $\mathbf{0.7927} \pm 0.0186$ |

*The standard deviations for all ablation studies are obtained by testing on 2000 testing molecules.

## 4 Reproducibility statement

We did not introduce any task-specific neural network archiecture. The results presented in this study can be straightforwardly reproduced using publically available datasets and ready-to-use implementation of convolution and Transformer from either `PyTorch` or `TensorFlow`. We have also provided detail hyper-parameter setup to ensure reproducibility. We have included the complete code for reproducing the conformation generation results in `https://anonymous.4open.science/r/TensorVAE-4576/` and code for reproducing the property prediction results in `https://anonymous.4open.science/r/TensorVAE-0DE7`.

## 5 Conclusion

We develop TensorVAE, a simple yet powerful model able to generate 3D conformation directly from a 2D molecular graph. Unlike many existing work focusing on designing complex neural network structure, we focus on developing novel input feature engineering techniques. We decompose these techniques into three main ideas, and explain how one idea naturally connects to the next. We first propose a tensor representation of a molecular graph. Then, we demonstrate that sliding a rectangle kernel through this tensor in an 1D convolution manner can achieve a global information aggregation. Finally, we present the complete CVAE-based framework featuring 2 transformer-based encoders and another transformer-based decoder, and propose a novel modification to the first multi-head attention layer of the decoder to enable sensible integration of the output of the other two encoders.

The proposed TensorVAE demonstrates state-of-the-art generative performance compared to recently proposed deep-learning-based generative models on the GEOM dataset, utilizing DFT-generated unbound conformations. When directly applied to the Platinum dataset, which contains ligand-protein bound conformations, the proposed method offers faster generation speed while maintaining competitive accuracy as compared to 5 popular and classical methods.

**Limitations and Future Directions**. Despite achieving promising performance in conformational generation, the current work has three major limitations that pave the way for future improvements. Firstly, the proposed tensor graph representation lacks invariance under random permutations of atom ordering. While experimentally robust, achieving true invariance to such transformations would enhance the stability of conformation generation.

Secondly, the training process only achieves approximate SE(3) invariance due to the presence of $R$ in the input of the posterior encoder. Aiming for exact invariance has the potential to further improve the TensorVAE framework's performance. To address this, we plan to replace $q_w(z|R, G)$ with the recently proposed equivariant posterior encoder component from the Geometric AutoEncoder framework in GeoLDM (Xu et al., 2023).

Thirdly, the current TensorVAE can only predict the local structure of molecules, specifically heavy atom coordinates with respect to an arbitrary origin. It lacks the capability to predict the SE(3) transformation necessary to obtain the bounding pose with respect to a protein target—a crucial aspect for structural-based drug discovery tasks. Our next objective involves expanding TensorVAE's architecture to predict both unbound ligand and protein conformers as input and produce valid bound ligand conformation as output. To achieve this, we are integrating the SE(3) equivariant convolution operation proposed in the TensorField Networks (Thomas et al., 2018) into the TensorVAE framework. This expansion aims to enhance the model's ability to generate conformer ensembles suitable for docking to specific protein targets.

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

## A   Appendix

### A.1   Global information aggregation beyond the $N^{th}$-hop

A geometric interpretation of GNN's message passing layer is it aggregates information between atoms (and their bond) that are 1-hop away. With $L$ layers, information from atoms that are $L$-hop apart can be aggregated. Here, we define a global information aggregation as the $N^{th}$-hop aggregation with $N$ being the total number of atoms, where each atom is able to aggregate information from all other atoms.

It is worth noting that for a fully-connected GNN, a 1-hop message passing can already achieve this global information aggregation. Transformer's self-attention can be considered as a type of fully-connected GNN. However, a vanilla transformer can only aggregate features from each token/atom; if edge features are not included, they needed to be incorporated somehow through additional inputs (e.g. the pair interaction matrix of Uni-Mol). The primary reason motivating the creation of the fully-connected tensor representation is we want each generated token contain both atom and bond features, such that we can eliminate the pair interaction or bond matrix. To achieve this, we fill each column of the fully-connected tensor with;

- focal atom features;

- chemical and virtual bond features indicating how the focal atom is connected to all other atoms;

- atom features of all connected atoms, since for off-diagonal cell, we sum atom features of both the neighbour atom and the focal atom.

Running a $N \times 1$ kernel filter on the proposed tensor also achieves a global information aggregation. By increasing kernel width to 3, the aggregation window includes global information from two immediate neighbours. This type of information aggregation extends far beyond just $N^{th}$-hop.

More interestingly, when multiple kernels are applied simultaneously to the same $N \times 3 \times C$ region, each kernel is free to choose whichever group of atom/bond features to attend to depending on its kernel weights. This resembles the multi-head attention mechanism of a transformer, where each kernel(head) contributes to a portion of the generated feature token. We believe the effective global information aggregation driven by these two (tensor representation + 1D Conv) simple yet intuitive ideas is the main reason why the proposed TensorVAE achieves superior performance with a much less number of parameters.

## A.2  Connection to a fully-connected Message Passing GNN

We show that information aggregation achieved by running a $1 \times 1$ convolution over the proposed tensor representation is similar to that achieved by a fully-connected MPNN (Gilmer et al., 2017).

When running a $1 \times 1$ convolutional operation over the proposed tensor, a $W \in R^{1 \times 1 \times F \times C}$ kernel matrix is shared among all $N \times N$ cells of the tensor, where $F$ is the number of kernels and $C$ is the channel depth. Since each cell, regardless it is on-diagonal or off-diagonal, is stacked with an atom feature vector and a bond feature vector, the weight matrix can be decomposed into two parts, $W_v \in R^{F \times C_v}$ and $W_e \in R^{F \times C_e}$, where $C_v$ is the atom feature vector size and $C_e$ is the bond feature vector size. The bond feature vector for on-diagonal cells is filled with zeros, since there is no self connection for focal atoms. Subsequently, for each column $n$ of the tensor, a $1 \times 1$ convolution operation followed by a sum-aggregation over the rows can be decomposed into 3 steps;

- **Off-diagonal cell aggregation**. For each off-diagonal cell, we first sum the atom feature vectors of the focal atom and its cell-specific neighbour atom, as described in Fig.1. Due to convolution operation, the dot product of the summed vector and $W_v$ is then computed. Simultaneously, the dot product between the bond feature vector and $W_e$ is also computed. The resulting two feature vectors are added together. This aggregation process can be expressed as;

$$W_v h_n^0 + W_v h_m^0 + W_e e_{n,m} \in R^{F \times 1}$$

  where $h_n^0 \in R^{C_v \times 1}$, $h_m^0 \in R^{C_v \times 1}$ and $e_{n,m} \in R^{C_e \times 1}$ are the focal atom feature of the $n^{th}$ column, neighbour atom feature of the $m^{th}$ cell in column $n$, and bond feature between the $n^{th}$ focal atom and its $m^{th}$ neighbour atom, respectively. If we concat $h_n^0, h_m^0$, and $e_{n,m}$ into a single vector $(h_n^0, h_m^0, e_{n,m})$[5], this operation can also be represented as;

$$M\left(h_n^0, h_m^0, e_{n,m}\right) = (W_v, W_v, W_e)^{\in R^{F \times (C + C_v)}} \cdot (h_n^0, h_m^0, e_{n,m})^{\in R^{(C + C_v) \times 1}}$$

- **Row-wise aggregation**. The above aggregation operation generates a feature vector (of size $R^{F \times 1}$) for each off-diagonal row of column $n$. The sum-aggregation over these rows generates a feature vector which contains the aggregated information from all neighbour atoms.

$$m_n^1 = \sum_{m \in N_{\setminus n}} M\left(h_n^0, h_m^0, e_{n,m}\right)$$

- **Complete aggregation**. Finally, we aggregate this feature vector $m_n^1$ onto the focal atom feature to complete the sum-aggregation operation over all the rows of column $n$.

---

[5]we define (●) as a concatenation operator as in MPNN (Gilmer et al., 2017)

$$h_n^1 = U\left(h_n^0, m_n^1\right) = \mathbf{ReLu}\left(W_v h_n^0 + m_n^1\right)$$

Noticeably, the $M$ and $U$ operators correspond exactly to the message passing phase of a single forward pass of a fully-connected MPNN, as described by Eqs.1 and 2 of the MPNN paper (Gilmer et al., 2017).

Similarly, the feature aggregation operation of a $N \times 1$ kernel can be expressed as;

$$h_n^1 = \mathbf{ReLu}\left(W_v^n h_n^0 + \sum_{m \in N_{\backslash n}} \left(W_v^m h_n^0 + W_v^m h_m^0 + W_e^m e_{n,m}\right)\right)$$

This type of aggregation is more flexible and has more expressive power as different node and edge features are weighted differently. This flexibility is further increased with a $N \times 3$ kernel whose corresponding aggregation can be expressed as;

$$h_j^1 = \mathbf{ReLu}\left(\sum_c^{(i,j,k)} W_v^{cc} h_c^0 + \sum_c^{(i,j,k)} \sum_m^{N_{\backslash c}} W_v^{cm} h_c^0 + W_v^{cm} h_{cm}^0 + W_e^{cm} e_{c,m}\right)$$

where $i, j, k$ are the indices of three adjacent columns. In this respect, **the information aggregation achieved by a fully-connected MPNN is a special case (the simplest form) of a more general framework embodied by a single convolution operation over the proposed tensor representation**.

### A.3    Training hyperparameters

**Conformation generation**. Training is conducted on a single Tesla V100 GPU. We follow a similar learning rate schedule, shown by Eq.3 of the original Transformer paper (Vaswani et al., 2017) but with $d_{model} = 9612$. This results in a maximum learning rate of $1.6e^{-4}$. To tackle the notorious issue of KL vanishing (Fu et al., 2019), we set a minimum KL weight of $1e^{-4}$ and double it every $62.5e^3$ iterations until a maximum weight of 0.0256 is reached. We select Adam optimizer (Kingma & Ba, 2015) default hyperparameters for training. We present some interesting observations of the training/validation curve corresponding to this setup in Sec.A.7. For both experiments, the TensorVAE is trained for $1e^6$ iterations with a batch size of 128. The implementation details of NaiveUNet is explained in Sec.A.4.

**Molecular property prediction**. For the molecular property prediction task, we use the same GDR transformer encoder structure (4 attention layers and approximately 5M parameters) and add an additional mean pooling layer, which is then followed by a linear layer for property prediction. We follow the same data train-val-test split in Uni-Mol and GEM and standardize the output property data. We train the GDR model for 300 epochs with a batch size of 128. The learning rate schedule is the same as that of the TensorVAE.

## A.4 NaiveUnet model architecture

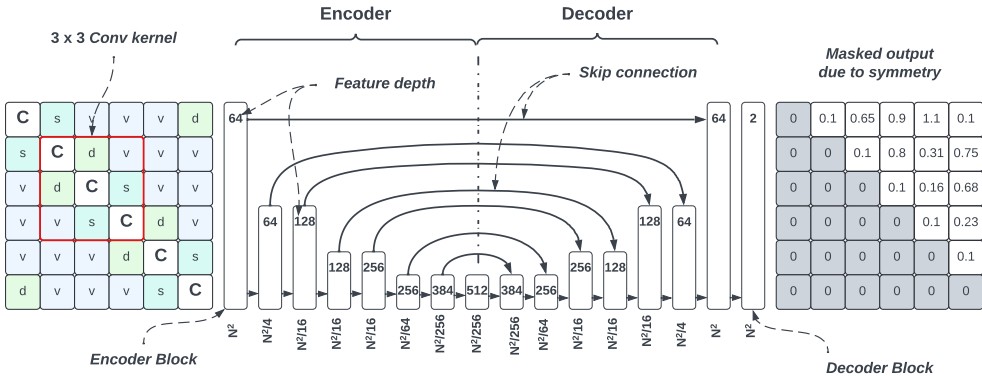

Figure 7: Naive UNet model. $N = 69$

We train the above NaiveUNet on the Drugs dataset for 30 epochs with a constant learning rate of $1e^{-4}$, and batch size of 32. We follow the same method presented in GraphDG (Simm & Hernández-Lobato, 2020) to convert the predicted distance matrix to conformation.

## A.5 Atom and bond features

We list the atom features and bond features together with the encoding method used to construct the proposed tensor in Tab.10.

Table 10: Atom and bond features used to construct input tensor.

| Feature name | Feature value | Encoding method |
|---|---|---|
| Atom type | H, C, N, O, F, S, Cl, Br, P, I, Na, B, Si, Se, K, Bi | one-hot |
| Atom charge | -2, -1, 0, 1, 2, 3 | one-hot |
| Atom chirality | Unspecified, Tetrahedral_CW Tetrahedral_CCW, Other | one-hot |
| Bond type | Single, Double, Triple, Aromatic, Virtual | one-hot |
| Normalized bond length | - | real-value |
| Bond stereochem | StereoNone, StereoAny, StereoZ StereoE, StereoCIS, StereoTrans | one-hot |
| Bond in-ring size | 3 - 10 | one-hot |
| Coordinate (3 channels) | - | real-value |
| Pair wise atom distance | - | real-value |

## A.6 COV and MAT precision results

The precision COV and MAT scores are defined as;

$$\text{COV}_P\left(\mathbb{C}_r, \mathbb{C}_g\right) = \frac{1}{|\mathbb{C}_g|} \left|\left\{\hat{R} \in \mathbb{C}_g | \text{RMSD}\left(R, \hat{R}\right) \leq \delta, \forall R \in \mathbb{C}_r\right\}\right|$$

$$\text{MAT}_P\left(\mathbb{C}_r, \mathbb{C}_g\right) = \frac{1}{|\mathbb{C}_g|} \sum_{\hat{R} \in \mathbb{C}_g} \min \text{RMSD}\left(R, \hat{R}\right)$$

Table 11: Precision performance comparison on GEOM Drugs dataset

| Method | $COV_P(\%) \uparrow$ | | $MAT_P(\text{Å}) \downarrow$ | |
|--------|------|--------|------|--------|
| | Mean | Median | Mean | Median |
| GraphDG | 2.08 | 0.00 | 2.4340 | 2.4100 |
| CGCF | 21.68 | 13.72 | 1.8571 | 1.8066 |
| ConfVAE | 22.96 | 14.05 | 1.8287 | 1.8159 |
| ConfGF | 23.42 | 15.52 | 1.7219 | 1.6863 |
| GeoDiff | 61.47 | 64.55 | 1.1712 | 1.1232 |
| TensorVAE[2] | **72.12** | **79.02** | **1.0655** | **1.0355** |
| | $\pm 1.5$ | $\pm 1.9$ | $\pm 0.0145$ | $\pm 0.0166$ |

*Results for GraphDG, CGCF, ConfVAE, ConfGF and GeoDiff are taken from (Xu et al., 2022).

## A.7 Training and validation curve

We present the train and validation plots for KL and reconstruction loss based on Drugs dataset in Fig.8a and Fig.8b, respectively. Both plots are based on an initial KL weight of $1e^{-4}$ doubling every $62.5k$ iterations (40 epochs). While KL validation loss reached 18.29 after $1e^6$ iterations (640 epochs), the reconstruction/RMSD loss reached 0.64Å at the end of training. During the first 5 epochs of training, model learning focused on reducing the KL loss due to it is orders of magnitude larger than the RMSD loss. We were expecting this trend to continue for a while until both losses converge roughly in the same range. However, much to our surprise, the model seemed to find a way to drastically reduce RMSD loss much earlier by leveraging the information from the GDR encoder; it learned to "cheat" by directly reversing coordinate information embedded in the output of GDR encoder back to the original conformation. The RMSD loss dropped to as low as 0.08Å. On the other hand, the KL loss climbed to almost 800, signaling signifcant divergence from standard normal distribution. At this stage, output of the GDR encoder contains informative features of the original 3D coordinates. With the KL loss weight increasing, it becomes more difficult for the model to cheat since training is forcing the output of GDR encoder to conform to a standard uninformative Gaussian distribution. The KL loss started to drop while the RMSD loss remained steady, indicating increasing reliance on the output of G encoder for reconstructing the conformation. As the output of GDR encoder becomes less informative, the model learned to rely almost entirely on the aggregated feature from the G encoder to decode conformation.

We attempted to initiate the training with a much larger initial KL weight ($1e^{-2}$) to prevent "cheating" from begining. However, this quickly led to the notorious KL vanishing issue (Fu et al., 2019). We figure that "cheating" is actually beneficial in that it reduces learning difficulty particularly for the decoder; its weights are tuned on easy training task, simply reversing what GDR encoder has done. In other words, the tuned weights of the decoder already hold crucial information on how to decode highly informative input features. As KL weight increases, model learning shifts to make the output of G encoder more informative. Also, this maybe an easier learning task as the RMSD loss is already very low (back-propagation of this loss contributes little to weight update); instead, model learning primarily focuses on optimizing the KL loss. This two-stage iterative loss optimization is much easier than optimizing both losses simultaneously throughout the training process.

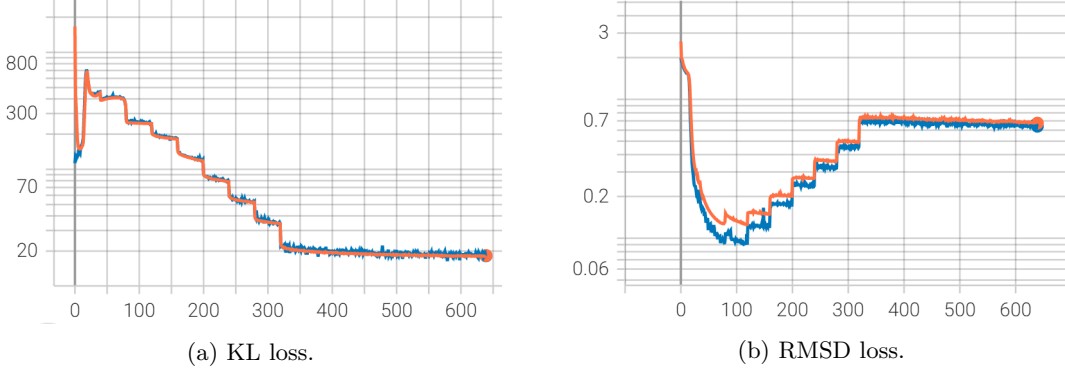

(a) KL loss.                    (b) RMSD loss.

Figure 8: Training and validation plots for Drugs dataset. Orange line: Train; Blue line: Validation

### A.8 Molecular property prediction

Following Uni-Mol (Zhou et al., 2023) and GEM (Fang et al., 2022), we report property prediction result on the MolecularNet (Wu et al., 2018) QM9 regression task. The goal of this task is to estimate *homo*, *lumo*, and *homo-lumo gap* properties of molecules in the QM9 dataset based on their molecular structure. We adapt the proposed GDR encoder to this regression task by changing its prediction head. We defer the details of this adaption and training procedure to SecA.3. We report the mean average error(MAE) over all the test samples.

The result of the adapted model is compared to those of 7 other models including;

- D-MPNN (Yang et al., 2019), AttentiveFP (Xiong et al., 2019) and GEM which are GNN based models without pretraining;

- N-Gram (Liu et al., 2019), PretrainingGNN (Hu et al., 2020) and GROVER (Rong et al., 2020) with pretraining;

- a variant of Uni-Mol without pretraining.

The MAE for all compared methods are summaried in Tab.12. The proposed GDR encoder produces a SOTA performance with less than 5M parameters. This experiment demonstrates that the proposed feature engineering method is very effective at information aggregation.

Table 12: Property prediction result comparison based on MolecularNet QM9 benchmark.

| Method | MAE |
| --- | --- |
| D-MPNN | 0.00814 (0.00001) |
| AttentiveFP | 0.00812 (0.00001) |
| N-Gram | 0.00964 (0.00031) |
| PretrainGNN | 0.00922 (0.00004) |
| GROVER base | 0.00984 (0.00055) |
| GROVER large | 0.00986 (0.00025) |
| GEM | 0.00746 (0.00001) |
| Uni-Mol w/o pretraining | 0.00653 (0.00040) |
| GDR encoder (ours) | **0.00553** (0.00012) |

*All results are taken from (Zhou et al., 2023). Values in parenthesis are standard deviation obtained by repeating experiments 4 times.

