# OpenReview forum: "TensorVAE: a simple and efficient generative model for conditional molecular conformation generation"
_TMLR — Accepted by TMLR_

### Review · Reviewer_SZqZ · 2023-11-03

**Summary Of Contributions:**

This paper proposes TensorVAE: a VAE based on a specific tensor representation which can be used to both generate 3D conformations of molecules and predict their properties. The contributions are:

- The TensorVAE architecture and data format. However, I am uncertain to what degree this can be considered a contribution: TensorVAE seems to fall into the category of "specialized GNN" or "special case of transformer", but it is described in a somewhat unclear way that makes me uncertain about this equivalence.
-  A transformer-VAE hybrid with a modified attention (although again this is not very clearly described so its contribution is unclear)
- Experiments showing strong results for conformer generation

**Audience:**

No

**Claims And Evidence:**

Yes

**Requested Changes:**

I am not really sure what to write for this: TMLR asks for claims which are well-supported and of interest to the audience. I can't really decide what the key claims of the paper are. I would guess either:

- "TensorVAE is a great method for predicting conformers". This claim seems fairly well-supported, but given that TensorVAE seems to just be a transformer variant, I'm not sure how interesting such a claim is: have the authors essentially just tuned a transformer?
- "The featurization of TensorVAE somehow really improves upon previous work": this would be an interesting claim but it is not well-supported, since when comparing to other methods the authors are changing not just the featurization but also the model, so it is not clear what is actually causing the performance.

Ultimately my request to the authors is to clarify what the key claims are for the paper and explain how they are supported. For this reason I have written "claims not supported" in my review below.

Also, it is not clear that the method is actually equivariant as the authors suggest: when describing equation 3, the authors say that the encoder "does not involve any coordinate", implying it is equivariant. However, it appears that it *does* actually involve coordinates, as these are included in $R$, no?

**Strengths And Weaknesses:**

An overarching weakness of this paper is its clarity. Although I have certainly read less clear papers, many things were not described very clearly in this paper: particularly the model (e.g. how $p_\theta(R|z,G)$ decomposes into several sequential decoders and how the attention masking works). I think an explicit algorithmic statement of the model would help for clarity. This issue makes it difficult to properly assess the correctness and impact of the claims made in the paper. The following review content is therefore based on my best guess about what I think the authors are doing.

The next thing I would comment on is the model and featurization (the efficacy of which is a key claimed contribution from the authors). The proposed featurization seems to just be an adjacency matrix with features, differing from a standard MPNN only in that the atom and edge features are explicitly mixed rather than being maintained separately. This seemed like a fairly small implementation detail to me. Next, the authors propose a kind of masked convolution, which seemed very similar to the attention mechanism of a transformer. In the majority of the paper the authors seem to refer to their model simply as a transformer, making me believe that it is just a transformer variant. The authors go on to use this transformer as the encoder/decoder of a VAE, which is not very novel (e.g. https://arxiv.org/abs/2207.13529).

Finally, I will focus on the experiments. The reported results of the experiments are quite strong, and if correct would be one of the strongest parts of the paper. However, it is not totally clear to me that these results are correct/comparable with previous work. The authors state that their method is "simple" and mainly relies on a good choice of features. It is therefore surprising that it outperforms a lot of other methods by such a large margin. My intuition is that the authors have made some kind of mistake here (I would bet ~3:1 odds). Possible sources of this are:
- Different random selection of train/test molecules (it was not clear if the exact same set of molecules from Xu et al 2022 was used or whether it was just generated with a similar same procedure)
- Data discarding: the authors seem to have discarded a fraction of the data which had too many atoms. Even though the fraction is small, presumably the other methods did not discard it. Maybe it has an outsized influence on the overall error?
- Subtle data leakage: for example, the authors use a depth-first traversal to order the nodes in the tensor. Maybe there is actually some spurious correlation here which the model picks up on?
- Some other small inconsistency (possible because the authors are copying results from previous papers rather than re-running things). E.g. a different normalization or different value of $\delta$ in equation 4.

I should clarify that I have not *found* a bug, just that I am highly suspicious of the results.

Summing up:

Strengths of the paper: a generally sensible method, good experimental results (if correct)

Weaknesses of the paper: lack of clarity, little novelty

---

> ### Author Response · Authors · 2023-12-12
> **Reply to reviewer SZqZ: Part 1**
>
> We would like to express our deep gratitude to reviewer SZqZ for your thorough review and constructive comments. We intend to address your concerns in three parts. In part 1, we will address the issue raised against our claims and their supporting evidence. In part 2, we will further clarify the experimental procedures we have followed and how we ensured a fair comparison with other methods. Finally, in part 3, we will summarize the novelty of proposed TensorVAE.
>
> **Q1 Claims and Evidence**
>
> We have dedicated the last paragraph of the introduction to summarize the three main claims/contributions of the proposed work. We reiterate them here again--we have proposed a conformation generation method that is: ***1. direct and efficient to run***; ***2. simple*** as it only replies on a standard convolution operation and *off-the-shelf* transformer encoder architecture; ***3. easy to implement*** as standard deep learning libraries including TensorFlow and PyTorch have provided ready-to-use implementation of these two fundamental building blocks.
>
> To substantiate the above claims, we have conducted extensive experiments and shown that our proposed method achieved promising performance with a much less number of parameters (**10X** less than DMCG and **7X** less than Uni-Mol) as compared to two recently proposed GNN and Transformer based methods: namely DMCG and Uni-Mol; both of which require sophisticated modification of the original GNN and or transformer architecture. Further, both methods also necessitate complex postprocessing step in addition to the standard *Kabsch Alignment* for loss computation. While DMCG requires an exhaustive search to obtain a collection of all permutation operations on symmetric atoms, Uni-Mol requires an additional SE(3)-equivariant head to convert pairwise positional offsets to 3D coordinates. On the other hand, we did not introduce significant change to the transformer architecture except for a modified self-attention mechanism that merges the output of the two encoders (G and GDR encoders). In terms of loss function, we only use the Kabsch aligned loss.
>
> As compared to the recently proposed diffusion-based method GeoDiff, our method does not require sophisticated design and customized implementation of SE(3) equivariant transition kernel. It is also a direct generative model that avoids inefficient sequential sampling steps to generate conformation. Our method not only outperforms GeoDiff by a large margin, it is also about 100X faster.
>
> To demonstrate ease of implementation and ensure reproducibility, we have released all our codes to show how we leverage TensorFlow's standard transformer implementation to build the TensorVAE model. In the appendix, we have made every possible effort to ensure our results are reproducible. This includes sharing the training and validation curve to demonstrate the training stability of the model and a detailed explanation of the training rate and KL weight schedule to ensure that anyone running our code will see the exact same training and validation curve.
>
> **Q2. lack of support for performance gain from feature engineering**
>
> We'd like to draw a distinction between featurization and feature engineering. Our featurization method largely overlaps with those of the compared baselines, including atom features, bond features, 1-hot encoding, using pairwise distance, etc. In fact, this is intentional as we aim to show that the performance gain of the proposed TensorVAE is not a result of different featurization but rather is attributed to novel feature engineering. Our feature engineering consists of 3 main ideas as clearly outlined in the original manuscript. The first 2 focus on the utilization of tensor representation followed by an 1D convolution operation to achieve a global information aggregation among atoms and bonds in a molecule. We have explained in the Appendices A1 and A2 that such feature engineering step achieves a more comprehensive information aggregation as compared to that of MPNN which is commonly adopted in other compared baselines.
>
> Subsequently, we conducted 4 ablation studies in Appendix A8 (now in section 3.3 of the main text) which substantiate why the unique combination of tensor representation and 1D convolution is crucial to achieving a promising performance.
>
> The last feature engineering method tackles a training instability issue caused by the randomness in the sampled latent variables when they are used directly to compute attention weights. The solution is only using them as value matrix and compute Query and Key matrices (which in turn used to compute attention weights) exclusively from the output of $\sigma_{\theta_1}$. This leads to stable training process as shown in Appendix A7. Otherwise, the training and validation losses fluctuated randomly and did not converge at all in our experiment.
>
> In conclusion, the performance gain from all 3 feature engineering methods are well supported by experimental results.

---

> > ### Comment · Reviewer_SZqZ · 2023-12-21
> > **Featurization vs feature engineering**
> >
> > You state:
> >
> > > We'd like to draw a distinction between featurization and feature engineering.
> >
> > What in your opinion is the difference between these terms? This was not clear from your response. I personally consider them to be synonyms.
> >
> > If you are referring to things like the 1D convolution as "feature engineering" I think this is not what most people in the field mean by this term. I think this is usually called the "model architecture" or something similar.

---

> > ### Comment · Reviewer_SZqZ · 2023-12-21
> > **Are you sure these are the claims you wish to make?**
> >
> > I did read your paper and saw the 3 claims at the end of the introduction. The reason I thought I misunderstood the claims is because these claims don't involve performance and are therefore in my opinion completely uninteresting (and also a little bit vague). Claim 1 basically says it generates conformations from a graph without some expensive internal computation (but what constitutes "a single step" is not totally clear). Is this not mainly a claim about the input and output types? Claims 2-3 basically say that the method is simple. Critically, none of these claims involve the performance of the method! I think a method which generates random coordinates for each input atoms would also satisfy these claims while being more efficient, simpler, and easier to implement. This is why I tried to find additional implicit claims about performance in my review.
> >
> > Ultimately though, if the authors wish for their paper to be evaluated according to these claims then I will do so. I changed my review to say that the claims are well supported (because they are trivially true) but not interesting to TMLR's audience (because any number of useless and uninteresting algorithms would satisfy these simple claims). This is probably not what the authors want though. I am happy to revise my review if the authors change their claims.

---

> ### Author Response · Authors · 2023-12-12
> **Reply to reviewer SZqZ: Part 2**
>
> **Q1. Different random selection of train/test molecules from Xu et al 2022 was used and Data discarding**
>
> As we have mentioned in the Dataset paragraph on page 7 of the manuscript, our method and all compared baselines have followed the same data generation procedure outlined in ConfGF[1]; that is selecting 200,000 conformations (top 5 conformations from 40,000 randomly selected molecules) for training, and then testing on 200 randomly selected molecules satisfying the following filtering conditions;
> 1. For GEOM-QM9 experiment, molecules with more than 50 but less than 500 annotated conformations.
> 2. For GEOM-DRUGS experiment, molecules with more than 50 but less than 100 annotated conformations.
>
> In the ConfGF paper, the total numbers of annotated conformations for 200 testing molecules are 22408 and 14324 for QM9 and Drugs respectively. In our sample of 200 molecules, there are slightly more testing conformations (23079 and 14396 respectively). Additionally, we realized that 200 randomly sampled molecules are probably not enough to substantiate our results. So, we increased the test-set size to include 2000 molecules, totalling 280,000 testing conformations for QM9 experiment and 144,000 for DRUGS experiment. The testing results for 200 random molecules are denoted as TensorVAE1 in table 1, and the testing results for 2000 random molecules are denoted as TensorVAE2. Note that, for both cases, the TensorVAE model was the same which was trained only on 200,000 conformations. This clearly demonstrates the generalization capability of the model. In comparison, none of the compared baselines have conducted test of similar scale or have reported any standard deviation.
>
> To further address your concern, we have conducted testing on the **same set of 200 GEOM-DRUGS molecules** (downloaded from [here](https://drive.google.com/drive/folders/1b0kNBtck9VNrLRZxg6mckyVUpJA5rBHh)) which GeoDiff has used. We first checked and made sure that none of the 200 molecules is in our training dataset. We then conducted a basic analysis to **determine that the largest molecule in this dataset contains 71 heavy atoms, only 2 atoms more than the largest (69) we have considered**.  Also, there are only 3 molecules having more than 69 atoms. For these 3 molecules that the current trained TensorVAE cannot process (but can easily handle it if we increase the max number of atoms to 71 and completely retrain the model. However, we don't consider this necessary since the majority of molecules consist of fewer than 69 atoms. Increasing the maximum number of atoms by 2 is unlikely to bring about significant changes in the results.), we consider the worst case scenario that the trained TensorVAE cannot produce any meaningful conformation and **their MAT scores are set to 2 $\overset{\circ}{A}$, and their COV scores are set to 0%**, and then we compute the performance stats as before. The results are reported below.
> 1. Without MMFF optimization
>
> | Models| COV mean$(\\%) \uparrow $ | COV median$(\\%) \uparrow$ | MAT mean $(\overset{\circ}{A})$ $\downarrow$ | MAT median$(\overset{\circ}{A})$ $\downarrow$ |
> | -------- | ------- | ------- | ------- | ------- |
> | GeoDiff | 88.45 |97.09 |0.8651 |0.8598   |
> | TensorVAE   | 93.05 | 98.98 | 0.8087| 0.7866|
>
> 2. With MMFF optimization
>
> | Models| COV mean$(\\%) \uparrow $ | COV median$(\\%) \uparrow$ | MAT mean $(\overset{\circ}{A})$ $\downarrow$ | MAT median$(\overset{\circ}{A})$ $\downarrow$ |
> | -------- | ------- | ------- | ------- | ------- |
> | GeoDiff | 92.27 |100 |0.7618 |0.7340|
> | TensorVAE   | 93.36 | 98.18 | 0.7267| 0.7032|
>
> TensorVAE still shows clear performance advantage over GeoDiff even under this worst case scenario.
>
> **Q2. Subtle data leakage**
>
> As mentioned in the **Determining input tensor size and atom ordering** paragraph on page 7, we have adopted a **random depth-first traversal** of the molecular graph. So the ordering of atom is random for every molecule. In fact, the ordering is also random for the same molecule in each of the 10 runs we have conducted to obtain standard deviation in Table 1. So there is unlikely any  spurious correlation between atom ordering and the generative performance.
>
> **Q3. A different normalization or different value of $\delta$ in equation 4.**
>
> We have strictly followed the same procedure as outlined in ConfGF[1] for determining the $C_g, C_r$ and cut-off threshold $\delta$. $C_g$ is the number of generated conformations which is twice as much as the number of ground-truth conformations $C_r$ per molecule. $\delta$ is set to 0.5$\overset{\circ}{A}$ for QM9 and 1.25$\overset{\circ}{A}$ for DRUGS. These settings are consistent across all compared baselines.
>
>
> [1] Chence Shi, Shitong Luo,Minkai Xu, and Jian Tang. Learning gradient fields for molecular conformation generation. In International Conference on Machine Learning, pp.9558–9568. PMLR, 2021.

---

> > ### Comment · Reviewer_SZqZ · 2023-12-21
> > **Improved experiments**
> >
> > I appreciate that the authors performed experiments on a larger test set. I think this is an improvement, but I have a few more questions about Table 1:
> >
> > - Why do error bars increase when you increase from 200 to 2000 test points? Should they not decrease by a factor of $\sqrt{10}$? Or are you reporting standard deviation rather than standard error?
> > - Is the test set the same between TensorVAE and the baseline? If not, then it is unclear whether the results are directly comparable. I think the Table deserves a more precise caption than "performance"

---

> ### Author Response · Authors · 2023-12-12
> **Reply to reviewer SZqZ: Part 3**
>
> In the final part of response, we first address 2 technical issues relating to the neural network architecture and then wrap up the response with a summary of novelty of the proposed method.
>
> **Q1. how $p_{\theta_2}(R|z,\sigma_{\theta_1}(G))$ decomposes into several sequential decoders and how the attention masking works**
>
> The architectural layout of the proposed TensorVAE follows the standard Conditional-VAE framework and is presented in Figure 3 of the manuscript. There are three main components: 1. a graph encoder ($\sigma_{\theta_1}$), 2. a posterior encoder ($q_w(z|R,G)$) and 3. the decoder for producing conformation  $p_{\theta_2}(R|z,\sigma_{\theta_1}(G))$. All three components share the exact same standard transformer encoder architecture: all have 4 layers (or encoder blocks), 8 heads and a latent dimension of 256. So $p_{\theta_2}(R|z,\sigma_{\theta_1}(G))$ does not decompose into several sequential decoders, rather it is simply a stack of 4 standard transformer encoder blocks as clearly shown in Figure 3.
>
> We did not introduce any architectural and operational change to these transformer blocks. This means that the attention masking works exactly the same as how it is working when training a standard transformer. Specifically, you can find out how we implemented masking operation in the `src.embed_utils.py` module of the provided code or you may instead refer to the official TensorFlow [tutorial](https://www.tensorflow.org/text/tutorials/transformer) for a more interactive explanation.
>
> **Q2. It is not clear that the method is actually equivariant as the authors suggest**
>
> We have claimed that the loss function presented in equation 2 is **roto-translational invariant**. The function has two main loss components: namely a reconstruction loss $$\log p_{\theta}(R|z,G) = \Sigma_{i=1}^{N}\Sigma_{j=1}^{3}(R_{ij}-A(\hat R,R)_{ij})$$
>
> and a regularization loss
>
> $$D_{KL}[q_w(z|R,G)||p(z)]$$
>
> The Kabsch alignment function $A(\hat R,R)$ aligns (through a rigid-body transformation) the predicted conformation $\hat R$ onto the ground-truth before loss computation. Consequently, this reconstruction loss should be the same regardless of the orientation (rotation) and translation of the predicted conformation with respect to the ground truth R. Therefore the reconstruction loss is roto-translation invariant.
>
> On the other hand, the regularization loss minimizes the KL divergence between the posterior $q_w(z|R,G)$ and a standard normal distribution $p(z)$. Upon convergence, we can reasonably assume that $q_w(z|R,G)$ approximates $p(z)$. So regardless of the orientation of R, $q_w(z|R,G)$ is always forced to conform to $p(z)$ which does not involve any R. Therefore, the regularization loss is also roto-translation invariant. Finally, for inference, we directly sample z from $p(z)$ rather than from $q_w(z|R,G)$ since the ground truth R is not available. This also removes the dependence on R. So in both training and inference, the proposed TensorVAE enjoys the roto-translational invariance.
>
> **Novelty of the proposed method**
>
> We argue that the novelty of the proposed TensorVAE lies in its simplicity and efficiency. These two properties are especially important for the practical application of the proposed method in structural High throughput virtual screening of drug candidates. The simplicity and efficiency are achieved not through ground-up new invention of novel architecture or sophisticated modification of existing paradigms, but rather through a combination of simple yet carefully designed and coupled feature engineering techniques that maximize the potential of simple existing model architectures on the molecular conformation generation task. The resulting capable model is simple to understand, easy to implement and much more efficient to run than other more sophisticated baselines.

---

> ### Author Response · Authors · 2023-12-21
> **Difference between featurization and feature engineering**
>
> Thank you again for being rigorous on this definition.
>
> In this context, we interpret featurization as the process of extracting relevant atom features (atom type, charge, chirality, etc.) and bond features (bond type, in-ring size, etc.) along with their rudimentary encoding methods (one-hot or real-valued) from a raw molecular graph, typically represented by the standard Mol object used by RDKit.
>
> To provide more clarity, we outline our featurization method in Tab.10 under Sec A5, where we detail the selected atom and bond features and their respective representation methods (one-hot or real-valued). It is worth noting that it is not uncommon to refer to this process as featurization, as seen in other works like DiffDock[1], ConfVAE[2] and GeoMol[3], where the extraction of atom and bond features, along with their encoding methods, is similarly labeled as featurization. This featurization process is almost identical for all other baselines and ours.
>
> Conversely, feature engineering involves additional processing of these selected features and their representations. This processing aims to create aggregated feature vectors (or input) that can be directly consumed by the standard transformer encoder module.
>
> The reason we'd like to draw this difference is to emphasize that the performance gain is not from different featurization but rather from how we engineer these features using simple representation and common operation such that a better performance can be achieved by using a standard transformer encoder architecture. This sets our approach apart from others, where achieving competitive performance often requires complex modifications to a standard architecture.
>
> [1] Corso, Gabriele, et al. "DiffDock: Diffusion Steps, Twists, and Turns for Molecular Docking." International Conference on Learning Representations (ICLR 2023). 2023.
>
> [2] Xu, Minkai, et al. "An end-to-end framework for molecular conformation generation via bilevel programming." International Conference on Machine Learning. PMLR, 2021.
>
> [3] Ganea, Octavian, et al. "Geomol: Torsional geometric generation of molecular 3d conformer ensembles." Advances in Neural Information Processing Systems 34 (2021): 13757-13769.

---

> ### Author Response · Authors · 2023-12-21
> **Clarification on our main claim**
>
> We are grateful for reviewer SZqZ for pointing out the ambiguity in our claims.
>
> What we really want to claim is actually already stated in the last sentence of the abstract as below
>
> **We show through experiments on two benchmark datasets that with intuitive feature engineering, a relatively simple and standard model can provide promising generative capability outperforming more than a dozen state-of-the-art models employing more sophisticated and specialized generative architecture.**
>
> We have also reiterated the above claim in the last paragraph of the introduction section by explicitly mentioning the three distinctive advantages which make the proposed TensorVAE more attractive than other baselines. In the same paragraph, we also mention that **We demonstrate through extensive experiments on two benchmark datasets that the proposed TensorVAE, despite its simplicity, can perform competitively against 22 recent state-of-the-art methods for conformation generation and molecular property prediction.** which essentially echoes the main claim in the abstract.
>
> In both places, we have linked the performance aspect of the proposed model to our claim.
>
> Please let us know if the above claim is still ambiguous, we are happy to make further adjustment to make it more clear.

---

> > ### Comment · Reviewer_SZqZ · 2023-12-22
> > **Claims make sense, basically performance is a fourth claim**
> >
> > Thanks for replying to this. I think we are "on the same page" about claims. It seems like TensorVAE's competitive performance with other methods is essentially a fourth claim. Do you think it would be helpful to add this to the numbered list in the introduction?

---

> ### Author Response · Authors · 2023-12-21
> **further clarification on experimental data**
>
> Thank you for your comment.
>
> Yes, we are reporting the standard deviation, and this is explicitly mentioned in the first paragraph above Tab 1 on page 9. To prevent similar confusion in the future, we will also include this statement in the footnote of Tab.1.
>
> We followed the exact training data generation procedure outlined in ConfGF[1] and GeoDiff[2], which is also adopted by other baselines. Regarding GeoDiff, its official github [repo](https://github.com/MinkaiXu/GeoDiff) implementation also provides an option to generate train/val/test using the same data generation procedure in ConfGF[1].
>
> On our end, we adhered to the exact procedure for generating our set of training, validation, and testing data for training the proposed TensorVAE. To demonstrate the generalization capability of our model, we tested it on a much larger dataset and obtained very consistent results. We anticipate that others have also conducted similar due diligence, as all their codes are released for public scrutiny.
>
> Furthermore, in our previous response to your comment regarding the fairness of the comparison (response link here https://openreview.net/forum?id=rQqzt4gYcc&noteId=tXHVR1Llz9), we repeated the experiment with the same set of testing data used by ConfGF, GeoDiff, and all other baselines. Our model continues to outperform all other baselines.
>
> [1] Chence Shi, Shitong Luo,Minkai Xu, and Jian Tang. Learning gradient fields for molecular conformation generation. In International Conference on Machine Learning, pp.9558–9568. PMLR, 2021.
>
> [2] Xu, Minkai, et al. "GeoDiff: A Geometric Diffusion Model for Molecular Conformation Generation." International Conference on Learning Representations. 2021.

---

> > ### Comment · Reviewer_SZqZ · 2023-12-22
> > **Clarification of dataset**
> >
> > Thanks for your response. I think this looks good. I suggest you clarify in section 3.3 which results are run on the *same* test set and which results are run on a test set generated with the same procedure. This will make it clearer to future researchers exactly which experiments were performed if they want to compare with your work in the future.

---

> ### Author Response · Authors · 2023-12-21
> **Clarification on single step conformation generation**
>
> We apologize for any confusion in explaining the concept of "single-step" generation. Let us clarify this aspect.
>
> When we refer to "single-step" generation, we are distinguishing it from two-step and distance-based methods (such as GraphDG and CGCF) as well as sequential-based methods (including GeoDiff, TorsionNet, and GFlowNet).
>
> Two-step distance-based methods involve the initial generation of pairwise distances between all heavy atoms, followed by the application of an Euclidean Distance Geometry algorithm to reconstruct atom coordinates from these distances.
>
> Sequential-based methods, like GeoDiff, iteratively refine a randomly generated set of atom coordinates through multiple sequential steps. Each step involves a single forward pass of the neural network architecture.
>
> In contrast to the aforementioned methods, our proposed TensorVAE takes a 2D molecular graph and a random latent sample as inputs during inference. It directly generates the conformational coordinates of all atoms in a single pass of the neural network structure. This eliminates the need for multi-step processes, making our approach more efficient and straightforward during the generation of conformational coordinates.

---

### Review · Reviewer_cqxm · 2023-11-04

**Summary Of Contributions:**

Summary:

In this submission, the authors propose a simple but effective method for conditional molecular conformation generation, which generates 3D molecular conformations from the corresponding 2D graphs. Technically, this work 1) proposes a tensor-based graph representation, encoding the atom and bond information jointly in a tensor format, and 2) applies 1D convolution to tokenize the tensor, leading to the following Transformer-based encoder. Therefore, the main technical contributions of this work are feature engineering and encoder architecture, in my opinion. Experimental results demonstrate the usefulness of the proposed method in several datasets.

**Audience:**

Yes

**Claims And Evidence:**

Yes

**Requested Changes:**

1. Each molecule may have various 3D conformations, is the proposed method able to generate various 3D conformations from a single 2D graph?

2. What is the full name of "GDR"?

3. It seems that the outputs of the decoder are 3D atom coordinates. Did the authors use other tools like OpenBabel to generate conformations from the coordinates?

4. When implementing the 1D convolution, a kernel with size N x 3 is applied. If my understanding is correct, N is the number of atoms in a molecule, which varies for different molecules. How to learn such a size-varying kernel?

**Strengths And Weaknesses:**

Strengths:

1. The paper is well-written and easy to follow. The key ideas and contributions are claimed clearly.

2. Many baselines are considered, and experimental results seem promising.

3. The method itself is simple and easy to reproduce.


Weaknesses:

1. If my understanding is correct, the 1D convolution applied to the graph tensor is not permutation-invariant, i.e., permuting the order of the rows and columns of the graph tensor leads to different token sequences. Accordingly, although the method is simple, I think it will lead to different 3D conformation coordinates when changing the order of atoms in the 2D graphs, which does harm to the rationality of the proposed method.

2. The proposed method seems to focus on generating conformations conditioned on 2D graphs. It might be more suitable to change the title to "conditional generation".

---

> ### Author Response · Authors · 2023-11-07
> **Reply to reviewer cqxm**
>
> We sincerely thank reviewer cqxm for your positive review and for providing constructive comments on our work. We would like to address your requested changes as following:
>
> First and foremost, we appreciate your title change suggestion and will add "conditional generation" in our title.
>
> **Q1. Is the proposed method able to generate multiple conformations per 2D graph**
>
> Yes. It is capable and is designed to do so. As outlined in the second paragraph on page 7 of the revised manuscript, titled **Direct conformation generation at inference time**, we first construct the G (2D graph) tensor based on the 2D molecular structure of a molecule. The G tensor then passes through the 1D convolution layer followed by $\sigma_{\theta_1}$ (4 layers of transformer encoder blocks) to produce $\left(h_1^L,...,h_N^L \right)$. Subsequently, we generate **a single latent sample** $(z_1,...,z_N)$ which is then combined with $\left(h_1^L,...,h_N^L \right)$ via the modified self attention to become the input of $p_{\theta_2}(R|z,h^L)$ (another 4 layer transformer encoder) for producing a **single** conformation (a set of 3D coordinates) R. **This method can be generalized to multiple conformation generation by sampling multiple latent samples for the same G tensor**. In actual implementation, if we want to generate 50 conformations for the same molecule, we obtain $\left(h_1^L,...,h_N^L \right)$ for a molecular graph and `np.tile` it 50 times; thereafter, we also generate 50 different latent samples using `np.random.normal(loc= 0, scale= 1)`. Then these 50 samples together with the duplicated inputs are batched and run through the proposed decoder $p_{\theta_2}(R|z,h^L)$ to generate 50 conformation in one go. Please see `get_prediction()` function in the `test_conf_gen.py` module in the provided [repository](https://anonymous.4open.science/r/TensorVAE-4576/) for our implementation of multiple conformation generation.
>
> **Q2. What is the full name for GDR tensor**
>
> There are two input tensors for training the proposed model, including G tensor and GDR tensor. The G tensor is the abbreviation for Graph Tensor as it only contains the 2D molecular graph information of a molecule. On the other hand, the GDR tensor is constructed by appending 4 additional channels to the G tensor. These 4 additional channels include "a distance channel" which holds all pairwise distances between atoms, and "3 coordinate channels" which holds the $(x, y, z)$ coordinates of all atoms. Therefore, GDR is an abbreviation for the tuple (Graph tensor, distance (D), coordinate (R)).
>
> Notice that for inference, only the G tensor and random latent samples are used as the input for generating conformations directly.
>
> **Q3. How is conformation generated from the produced 3D coordinates**
>
> The method for generating conformation from 3D coordinates is implemented in the `get_conformation_samples` function of the `test_conf_gen.py` module in the provided code. More specifically, once coordinates are generated, we embed the coordinates in the `rdkit` mol object using `GetConformer(0).SetAtomPosition` function. To visualize the embedded conformation, we use the `py3Dmol` package. The conformation visualization code is implemented in the `visualise_gen_mol.ipynb` in the provided code
>
> **Q4. How to train kernel weights with varying kernel size**
>
> The kernel size is fixed for all molecules in this work. As mentioned in the **Determining input tensor size and atom ordering** paragraph on page 7, we conducted a basic data analysis on both GEOM QM9 and DRUGS datasets to determine the appropriate input tensor size and the kernel height N. For the QM9 dataset, we iterated through all molecules and determined the maximum number of atoms to be 29. Subsequently, we set the input tensor height and width, and the kernel height to be 30. For the DRUGS dataset, we set the kernel height to be 69 which is the $98.5^{th}$ percentile of the number of atoms.
>
> For those molecules that have less than 69 number of atoms, we mask out the attention weights, KL losses and coordinate losses from those extra atoms. You can see how masking is implemented in the `src.embed_utils.py` module of the provided code.
>
> As outlined in the "Dataset" section on the same page, we have strictly followed the training molecule filtering process of [1] which only considers molecule in the DRUG dataset with more than 50 but less than 100 conformations for training and testing. This is also the same data generation procedure followed by all other baselines. With this filtering condition, the percentage of molecules satisfising the above filtering condition and having more than 69 atoms is only $0.19$% (69 covers more than 99% of all molecules after number of conformations filtering).
>
> [1] Chence Shi, Shitong Luo, Minkai Xu, and Jian Tang. Learning gradient fields for molecular conformation generation. In International Conference on Machine Learning, pp. 9558–9568. PMLR, 2021.

---

> ### Author Response · Authors · 2023-11-07
> **Addressing the second weakness raised by reviewer cqxm**
>
> We thank you again for your insightful comment.
>
> We have demonstrated experimentally that random permutation of the atom ordering will not affect the prediction performance of the proposed model.
>
> As outlined on page 7 in the paragraph "Determining input tensor size and atom ordering.", the ordering of the atoms along the diagonal of the input tensor is determined by a random start depth-first-traversal of the molecular graph for both training and prediction. You can see how we have implemented a random start depth-first-traversal atom ordering in the `mol_to_extended_graph` function of the `src.graph_utils.py` module of the provided repository.
>
> Additionally, in the TensorVAE1 experiment of table 1, we have run 10 experiments on the same set of 200 testing molecules, each with a different random ordering of the atoms per molecule. As indicated by the standard deviation of the result, random permutation of the atom ordering has very little effect on the prediction performance of the model.
>
> Finally, in the TensorVAE2 experiment, we expand the test set to 2000 molecules which is **10 X** more than the number of test molecules in all other baselines. We still do not observe significant performance variation between each test run. Therefore, we conclude that random permutation of the atom ordering does not affect the performance of the proposed method.

---

### Review · Reviewer_1HGp · 2023-12-11

**Summary Of Contributions:**

The paper proposes a new method for generating 3D molecular conformers from 2D graphs. The primary contribution of the method is encoding the input features of the 2D molecular graph as a tensor graph which includes atomic encodings (atom type, atom charge, atom chirality, atom coordinate) as well as bond encoding (bond type, bond stereochemistry, normalized bond length, distances) for all of the atoms in the system. In the tensor graph, the diagonal elements include the primary while the off-diagonal atoms represent information about the neighbors of the atom. Based on this tensor graph, the paper then proposes using a 1D-convolution as the primary neural network operation to process the tensor graph followed by a self-attention transformer. Taken together, the above make up the primary components of TensorVAE which is the paper's proposed method.

Next, the paper describes the formulation of the problem, including the VAE training loss and how it applies to the conformer generation case study and outline their primary experiments. The experiments focus on three main datasets: GEOM-QM9, GEOM-Drugs, and Platinum. The paper outlines the experimental conditions and metrics in detail before presenting the results on GEOM, where TensorVAE generally out performs the other methods the author compare against. The results include methods that apply force-fields for conformer generation in Table 2 and methods with and without force-fields in Table 1. In Table 3, the paper claims that TensorVAE outperforms other method while having less parameters and outline details about the compute efficiency during of TensorVAE compared to other methods.  Next, Table 4, Table 5 and Table 6 the paper provides an analysis of zero-shot performance of TensorVAE on the Platinum dataset, which includes molecular conformation in a protein environment. The results generally show reasonable performance on the Platinum dataset. Lastly, the authors provide a brief study on molecular property prediction along with a reproducibility statement and a conclusion.

**Audience:**

Yes

**Claims And Evidence:**

Yes

**Requested Changes:**

Important Requests that would sway my opinion:
* Please adjust the claims related to single-step generation and simple architectures. Are there other methods that perform conformer generation in a single step? How do they compare to TensorVAE.
    * You mention that your method does not require sophisticated neural network design, so it would be good to see performance with even simpler parts, such as MLPs.
* Add related work of sequence-based methods for conformer generation.
* Could you clarify why you only ran zero-shot experiments for Platinum? It seems like fine-tuning or training experiments for TensorVAE on Platinum would be appropriate.

Additional requests (nice to have):
* I would prefer seeing results of the ablation in the main text and would be OK moving the property prediction experiments to the appendix in exchange. Feel free to explain why you included property prediction in the main text.
* Could you clarify if the tensor graph representation is symmetric. If so, is there a way to take advantage of the symmetry? This could be interesting to discuss in the conclusion and future work section.
* It would be nice to expand potential future work directions.

**Strengths And Weaknesses:**

Strengths:
* The paper introduces a new representation for 2D molecular graphs that appears useful for conformer generation and additional tasks, such as property prediction.
* The paper provides an extensive set of experiments across relevant datasets (GEOM, Platinum) and compares with relevant methods. TensorVAE generally shows performance improvements and distinct advantages, such as better compute efficiency.
* The paper provides significant detail on the method and experimental settings.

Weaknesses:
* I think the claims of single step generation of conformers and simple architectures are somewhat overstated. Part of the text and the ablations in the appendix seem to indicate that neural network architecture does matter in performance, so it would be good to be clearer about the paper claims as important.
* It is unclear why the paper includes zero-shot experiments on Platinum only and why molecular property prediction experiments are also included in the main text. Given the focus on conformer generation, I think the results of the ablation would be more interesting to understand the important components of the proposed method.
* Related work is discussed primarily in the introduction and is missing a discussion on sequence-based methods for conformer generation, such as reinforcement learning [1] and GFlowNets [2].

[1] Gogineni, Tarun, et al. "Torsionnet: A reinforcement learning approach to sequential conformer search." Advances in Neural Information Processing Systems 33 (2020): 20142-20153.

[2] Volokhova, Alexandra, et al. "Towards equilibrium molecular conformation generation with GFlowNets." arXiv preprint arXiv:2310.14782 (2023).

---

> ### Author Response · Authors · 2023-12-16
> **Reply to reviewer 1HGp: Part 1**
>
> We sincerely thank reviewer 1HGp for your thorough review and helpful comments. We would like to address your requested changes as following:
>
> **Are there other methods that perform conformer generation in a single step? How do they compare to TensorVAE?**
>
> Yes there are. Among the 11 compared conformation generation baselines, 3 methods including CVGAE[1], DMCG[2], and Uni-Mol[3] are direct generative models producing conformation in a single pass of their corresponding NNs. Noticeably, both CVGAE and DMCG have adopted the same conditional VAE framework as we have done. Both these models utilize modified GNN/MPNN as their feature extraction backbone. On the other hand, Uni-Mol chooses a reconstruction framework which optimizes and refines an initial conformation generated by RDKit to arrive at the final conformation per molecule. The feature extraction backbone of Uni-Mol is a modified transformer architecture. As shown in Table 1 and Table 2, the proposed TensorVAE outperforms all the above 3 models.
>
> **You mention that your method does not require sophisticated neural network design, so it would be good to see performance with even simpler parts, such as MLPs**
>
> Thank you for this excellent suggestion. Simplicity, in our interpretation, refers to the absence of the necessity to introduce complex and task-specific modifications to any standard neural network architecture for domain-specific tasks. For instance, DMCG explicitly states its inability to achieve satisfactory performance using unmodified vanilla MPNN or GNN architectures. Instead, it relies on a combination of a modified GN block and GATv2 module, along with an additional modification allowing intermediate coordinate output per layer. Moreover, the loss function must account for all possible permutation symmetries in the molecular graph to achieve promising performance.
>
> Another instance is Uni-Mol, where the transformer architecture requires modification to consume pairwise representation matrices. Both models not only demand task-specific alterations to a standard architecture but also entail a significantly higher number of parameters to approach the performance level of ours.
>
> In summary, ***our main idea is to demonstrate that, through effective input feature engineering and the selection of a suitable foundational architecture (in our case, the transformer and Conditional VAE), there is no requirement for an inefficient multi-step process, intricate modifications, or a massive model size to achieve competitive performance***.
>
> Following your suggestion, we replaced the transformer encoder block with a MLP block of similar size and trained this variant for 100 epochs (lasting 3 days). We provide the detailed description of this MLP variant architecture in Sec 3.3 of the revised manuscript. We discovered that this MLP variant suffered from overfitting issue after the first 40 epochs, and it became worse after 90 epochs of training. We have added the training and validation curve of this variant in Sec 3.3 of the revised manuscript as well. Due to the worsening overfitting issue, we have decided it is not necessary to complete the training to show this variant has worse performance as compared to the proposed TensorVAE model.  In conjunction of the previous 4 ablation studies we have done, it seems effective information aggregation achieved by the proposed Tensor representation followed by an 1D convolution and the transformer attention mechanism both contribute to the superior performance of the proposed model.
>
> **Add related work of sequence-based methods for conformer generation.**
>
> We have added an dedicated paragraph (second paragraph on the second page of the revised manuscript) discussing the two suggested sequential based conformation generation models.
>
> [1] Mahmood, Seokho Kang, and Kyunghyun Cho. Molecular geometry prediction using a deep generative graph neural network.
>  scientific reports, 9(1):1–13, 2019.
>
> [2] Jinhua Zhu, Yingce Xia, Chang Liu, Lijun Wu, Shufang Xie, Yusong Wang, Tong Wang, Tao Qin, Wengang
> Zhou, Houqiang Li, Haiguang Liu, and Tie-Yan Liu. Direct molecular conformation generation. Transactions on Machine Learning Research, 2022. ISSN 2835-8856. URL https://openreview.net/forum?id=lCPOHiztuw.
>
> [3] Gengmo Zhou, Zhifeng Gao, Qiankun Ding, Hang Zheng, Hongteng Xu, Zhewei Wei, Linfeng Zhang, and
> Guolin Ke. Uni-mol: A universal 3d molecular representation learning framework. In The Eleventh
> International Conference on Learning Representations, 2023.

---

> > ### Comment · Reviewer_SZqZ · 2023-12-21
> > **MLP study is a good start but is not very well-done**
> >
> > I think the suggestion to compare with an MLP was good and appreciate that the authors started this experiment. However, it seems a bit poorly executed: the authors see to have just tried one architecture with regularization parameters like dropout set heuristically, and concluded that the transformer architecture was necessary based on this. I think such a claim would at the very least require tuning the regularization parameters to avoid overfitting rather than giving up after the first try, no? For example, one could turn up the dropout rate or applying $\ell_2$ regularization?

---

> > > ### Author Response · Authors · 2023-12-24
> > > **MLP ablation study with different dropout rates completed.**
> > >
> > > We thank reviewers 1HGp and SZqZ again for suggesting additional model runs with MLP as the backbone. We have finished running additional experiments with 3 dropout rates, including 0.3, 0.5 and 0.7. **We have added these additional results in Sec 3.3.**
> > >
> > > After training the model for 100 epochs for each dropout rate, we found that the dropout rate of 0.3 achieved the best validation KL and RMSD losses without encountering any overfitting issues. However, with an increase in the dropout rate, the RMSD training loss decreased (while the validation RMSD error remained the same), and the KL loss increased significantly. This behaviour mirrored that of the $1 \times 1$ convolution kernel (fully-connected MPNN) variant mentioned earlier. Essentially, the model increasingly relied on posterior encoder information to reconstruct the conformation and reduce the RMSD error, which is an easier task compared to reconstructing conformation from a 2D molecular graph in the absence of any coordinate information. This trend suggested that a higher dropout rate led to a reduction in the model's capacity to learn.
> > >
> > > Despite achieving the best performance among all tested dropout rates, after 120 epochs of training, the MLP variant with 0.3 dropout still performed significantly worse than the TensorVAE with a transformer backbone. Although its RMSD validation loss matched that of the TensorVAE, its KL validation loss was more than double that of the TensorVAE, indicating significantly lower learning capacity.
> > >
> > > We therefore concluded the attention mechanism is also a crucial contributing factor for effective information aggregation among atoms.
> > >
> > > We hope that the above response addresses your concern.

---

> > > > ### Comment · Reviewer_SZqZ · 2023-12-27
> > > > **New study looks much better**
> > > >
> > > > Thanks for these revisions. The new study looks much better. I will wait for other reviewers to respond before submitting my recommendation so that we can finish the discussion.

---

> > > > > ### Comment · Reviewer_1HGp · 2024-01-04
> > > > > **Thanks for the updated versions**
> > > > >
> > > > > I think that the revised version and discussion addressed most of my feedback. I have one "hard" edit request that I would like to see modified and one "soft" which would be nice to have but not necessary:
> > > > >
> > > > > - Hard Request: Please modify Figure 5 and Figure 6 to have better legends. I recommend having more describe names for the curves shown and removing the names of the ones not shown. The current version looks like it was taken from a screenshot of a software (e.g. Tensorboard) which is not preferred.
> > > > >
> > > > > - Soft Request: To further support the efficiency claim, it would be nice to have some additional data to support this, such as sample efficiency or compute time. I think the number of parameters in the model is already useful, but it could be further supplemented.

---

> > > > > > ### Author Response · Authors · 2024-01-06
> > > > > > **Acknowledging your request**
> > > > > >
> > > > > > We sincerely appreciate the further suggestions provided by reviewer 1HGp. In response to your request, we plan to address them as follows:
> > > > > >
> > > > > > 1. Hard request: We will revise figures 5 & 6 in the final version of the paper.
> > > > > >
> > > > > > 2. Soft request: In the current version of the paper, we have already presented (in Paragraph 1 on page 11) the inference time of TensorVAE for batch generating conformations for 200 DRUGS testing molecules (one molecule per batch, resulting in a total of 28,792 conformations, twice the number of reference conformations). The entire generation process took approximately 128 seconds, averaging 0.004 seconds per conformation. In comparison, for the same number of generated conformations, GeoDiff took 11,500 seconds, or 0.399 seconds per conformation. Notably, TensorVAE achieved this speed using a single core of the Xeon8163 CPU, while GeoDiff was executed on a single Tesla V100 GPU. The inference time of TensorVAE also includes the time required for converting an RDKit mol object to the proposed tensor graph.

---

> ### Author Response · Authors · 2023-12-16
> **Reply to reviewer 1HGp: Part 2**
>
> **I would prefer seeing results of the ablation in the main text and would be OK moving the property prediction experiments to the appendix in exchange. Feel free to explain why you included property prediction in the main text.**
>
> Thank you for your thoughtful suggestions. We have moved the ablation study (including the MLP ablation study) in the main text and moved molecular property prediction to the appendix.
>
> **Could you clarify why you only ran zero-shot experiments for Platinum? It seems like fine-tuning or training experiments for TensorVAE on Platinum would be appropriate.**
>
> The reason we conducted only zero-shot experiments is threefold. Firstly, the main purpose of the Platinum experiment is to demonstrate the generalization capability of the model rather than focusing on also achieving SOTA performance on it. The second reason is to ensure fairness of comparison. All other compared baselines in the Platinum experiment ran zero-shot prediction on all the 2912 conformations of the Platinum Diversified Dataset. It is only fair for us to do the same. Thirdly and most importantly, there is a better way to improve the performance on the Platinum dataset than simply finetuning TensorVAE on it. We provide the detailed explanation of this as following.
>
> In unbound vacuum conformation generation, since there is no fixed reference structure, a low RMSD is achievable by directly or indirectly predicting local molecular structures such as bond length, bond angles and torsion angles (or Dihedral Angles). The proposed TensorVAE and many compared baselines (GeoDiff, DMCG, Uni-Mol, etc) indirectly model these local structures by generating local coordinates of all heavy atoms at once with respect to an arbitrary origin. On the other hand, TorsionNet and GflowNets assume local structure such as bond angle and bond length are known (provided by RDkit's ETKDG method) and focus on only predicting torsion angles.
>
> After obtaining these local structures, we can use the Kabsch algorithm to compute an optimal rigid-body transformation, aligning the predicted conformation with the ground-truth for computing RMSD. In this regard, the loss function is SE(3) invariant. However, in the case of ligand-protein binding, the rigid-body transformation with respect to the protein structure, especially when binding to a protein pocket, also needs to be learned in addition to the local structure to correctly predict the bound conformation. Therefore, we believe that fine-tuning the proposed TensorVAE to achieve a better unbound RMSD does not add much value. In other words, the Platinum case study serves only to verify that TensorVAE can generalize to predict local structure correctly for the bound conformation generation task. This verification establishes a strong foundation upon which we can expand TensorVAE's model architecture so that it not only predicts SE(3) invariant local structure correctly but is also capable of predicting SE(3) equivariant rigid-body transformations.
>
>
> As mentioned in our conclusion, our primary focus for future development is the enhancement of the mentioned capabilities. Here, we would like to provide a brief expansion on this and share our main direction (along with the progress made). To enable TensorVAE to predict bound conformations, we are actively working on introducing an additional SE(3) equivariant component and prediction head for rotation and translation. Drawing inspiration from the Tensorfield Network[1], we aim to integrate the SE(3)-equivariant tensor product of representations into our network design. Progress has been made in expanding and decomposing the per-node feature ($h_i^l$) per transformer encoder layer into up to the $2^{nd}$ order irreducible representations.
>
> We are currently in the process of combining the expanded node representations (or equivalently $V_{acm}^{l}$ in [1]) with spherical harmonics (also up to $2^{nd}$ order) of the atom pairwise edge vectors using the Tensor Product operation. Once combined, the resulting tensor representations undergo further processing to produce SE(3) equivariant translation and rotation outputs. Despite the ongoing progress, there are still many other details we are working to figure out. One notable challenge is adapting the proposed modified self-attention mechanism in TensorVAE to be SE(3) equivariant as well. Despite these technical challenges, we remain dedicated to developing a model capable of producing bound conformations in a single step.
>
> **Could you clarify if the tensor graph representation is symmetric. If so, is there a way to take advantage of the symmetry?**
>
> Thank you for pointing this out. Yes, the tensor graph is symmetric. We do not yet have any concrete ideas on how to further exploit this property, but we will continue exploring this direction.
>
> [1]Thomas, Nathaniel, et al. "Tensor field networks: Rotation-and translation-equivariant neural networks for 3d point clouds." arXiv preprint arXiv:1802.08219 (2018).

---

> ### Author Response · Authors · 2023-12-21
> **Conducting more experiments with higher dropout rates**
>
> Again we thank reviewer SZqZ for your helpful suggestion.
>
> The design of the MLP block has been crafted to resemble the point-wise feedforward network found within the transformer encoder block. Both the MLP block and the point-wise feedforward network share a dropout rate of 0.1. Although we modified the last hidden layer size, indicated by the parameter `DFF` (you can find its value in the `src.CONSTS` module of the provided code), from 512 to 1024. This adjustment aims to ensure that the MLP variant possesses a similar number of parameters as TensorVAE, which totals around 11 million.
>
> The rationale behind increasing this hidden size is to underscore that the primary architectural difference between MLP and TensorVAE lies in the absence of the attention mechanism. Our intention is to demonstrate the crucial role of the attention mechanism in achieving competitive performance.
>
> We are also conducting experiments with 3 different dropout rates including 0.3, 0.5 and 0.7. We will report back our results back when they are ready (approximately within a week)

---

### Comment · Reviewer_SZqZ · 2023-12-21
**Authors' claims about invariance are likely not correct**

Hello, in the process of going through the reviews I noticed that reviewer cqxm and I both questioned the paper's claims about invariance. I thought it makes sense to post one comment for both these issues rather than post a separate comment for my own review.

I think that the authors are either using a different definition of invariance than the majority of researchers in the field or do not understand what invariance is. What I (and I believe the majority of researchers) mean by invariance is that an output does not change when inputs are modified by some operator (e.g. rotation or translation). This means not changing _at all_ (not even a little bit) and showing invariance generally requires a mathematical proof. I think that some invariance claims made by the authors clearly do not meet this criterion:

- By showing that the results have _very little_ (but not 0) variance when the atom permutation is changed in Table 1, the authors plainly demonstrate that their method is _not_ invariant with respect to permutation (despite stating the opposite conclusion in their response to reviewer cqxm).
- In their response to my doubt that the regularization loss is not invariant, the authors claimed that at convergence, $q(z|R,G)$ would approximate $p(z)$ (which does not depend on $R$). However, approximate does not mean _equal_. There is clearly a dependence on $R$, and if $R$ were ignored that would imply a VAE failure mode called "posterior collapse" and necessitate high reconstruction loss.

I think it would be more accurate to remove all claims about invariance from the paper. At best, I think the authors can claim to have "learned approximate equivariance". If the authors still wish to claim invariance, then I think a valid mathematical proof is required.

---

> ### Author Response · Authors · 2023-12-21
> **Agreeing to modify invariance claim**
>
> We extend our gratitude to Reviewer SZqZ for diligently scrutinizing our assertions to ensure their rigor.
>
> In terms of roto-translation invariance, we acknowledge Reviewer SZqZ's observation that, post-convergence of VAE loss, the neural network achieves only approximate invariance. Consequently, we will refrain from claiming roto-translation invariance and, instead, state that the model is capable of "achieving approximate invariance."
>
> In terms of permutation invariance, we have never claimed that the tensor representation followed by an 1D convolution is invariant under random permutation of the atom ordering. As mentioned in the first paragraph on page 9 of the revised manuscript, we claim that the model's generative performance, as measured by MAT and COV scores, **remains robust and does not significantly vary (or degrade) under random permutation of atom ordering**. This claim is experimentally validated, emphasizing the model's resilience to random permutations in atom ordering.

---

> > ### Comment · Reviewer_SZqZ · 2023-12-22
> > **Happy with response. Suggest you add a limitations section.**
> >
> > I am happy with these revised claims. I think lack of permutation invariance is ok, but I would personally like to see this described as a limitation of the method. The paper seems to currently lack any explicit discussion of the method's limitations. Perhaps it would be good to add this to the conclusion too?

---

> > > ### Author Response · Authors · 2023-12-22
> > > **Acknowledging your suggestions**
> > >
> > > We sincerely thank Reviewer SZqZ for patiently reviewing our response and consistently offering excellent suggestions.
> > >
> > > We are planning to address your suggestions as following:
> > >
> > > 1. Add a limitation section before the conclusion section to summarize the limitation of the model, including
> > > * Achieving approximate SE(3) invariance
> > > * Not permutation invariant
> > >
> > > 2. Add a fourth contribution point in the introduction specifically clarifying the performance claim of the proposed model
> > >
> > > 3. Explain clearly in Sec 3.3 the different test-set used to generate MAT and COV scores
> > >
> > > We are still in the process of running experiments with higher dropout rates. We will update you again once they are ready in about 3 days.

---

### Author Response · Authors · 2024-01-09
**Summary of changes made to manuscript**

We would like to extend our heartfelt gratitude to all reviewers for their invaluable contributions through critical assessments and constructive comments. Your insightful suggestions have significantly elevated the clarity of the revised manuscript. Below, we provide a summary of the major modifications made in response to your feedback:

**Title Modification**: We included "conditional generation" in the manuscript's title to better reflect the content.

**Additional Ablation Study**: An extra ablation study was incorporated, replacing the transformer architecture with an MLP architecture. We explored 4 MLP variants, with the dropout rate ranging from 0.1 to 0.7.

**Reorganization of Content**: The ablation studies have been relocated to the main text (Section 3.3), while the molecular property prediction case study has been moved to Appendix A8.

**Claim Modification**: On page 6 of the revised manuscript, we have fine-tuned the claim regarding "achieving roto-translation invariance" to "learning to achieve approximate roto-translation invariance" for enhanced precision. Furthermore, a paragraph has been added to summarize the limitations of the TensorVAE model, along with our proposed strategies to address them in the Conclusion section.

**Additional Contribution Claim**: In the last paragraph of the introduction section, a fourth contribution point was added, specifically elucidating the performance claim of the proposed model.

**Additional Results**: We introduced TensorVAE REF results, which are obtained on the same test dataset used in all other baselines. Detailed explanations regarding the distinctions between these test sets and results have been incorporated in Section 3.2.

**Figure formatting**: We have reproduced Figure 5 and Figure 6 with clearer images and improved legend presentation.

We trust that these comprehensive changes address your concerns, and we sincerely appreciate your time and efforts in reviewing our work.

---

> ### Comment · Reviewer_SZqZ · 2024-01-09
> **Changes are good**
>
> Thanks!

---

### Decision · Action_Editor_axW6 · 2024-01-22

**Recommendation:** Accept with minor revision

**Comment:**

This work proposes a method for conditional molecular conformation generation. After extensive engagement from the authors, all reviewers agree that the work meets TMLR's acceptance criteria. I thus recommend accepting the paper, and ask that the authors please do revise figures 5 and 6 as requested by reviewer 1HGp.

**Audience:**

Reviewers unanimously agree that the paper will be of interest to part of TMLR's audience.

**Claims And Evidence:**

Reviewers unanimously agree that the claims made in the paper are backed up by evidence.